# Structured Prediction Theory Based on Factor Graph Complexity

**Corinna Cortes**
Google Research
New York, NY 10011
corinna@google.com

**Vitaly Kuznetsov**
Google Research
New York, NY 10011
vitaly@cims.nyu.edu

**Mehryar Mohrii**
Courant Institute and Google
New York, NY 10012
mohri@cims.nyu.edu

**Scott Yang**
Courant Institute
New York, NY 10012
yangs@cims.nyu.edu

## Abstract

We present a general theoretical analysis of structured prediction with a series of new results. We give new data-dependent margin guarantees for structured prediction for a very wide family of loss functions and a general family of hypotheses, with an arbitrary factor graph decomposition. These are the tightest margin bounds known for both standard multi-class and general structured prediction problems. Our guarantees are expressed in terms of a data-dependent complexity measure, *factor graph complexity*, which we show can be estimated from data and bounded in terms of familiar quantities for several commonly used hypothesis sets along with a sparsity measure for features and graphs. Our proof techniques include generalizations of Talagrand's contraction lemma that can be of independent interest.

We further extend our theory by leveraging the principle of Voted Risk Minimization (VRM) and show that learning is possible even with complex factor graphs. We present new learning bounds for this advanced setting, which we use to design two new algorithms, *Voted Conditional Random Field* (VCRF) and *Voted Structured Boosting* (StructBoost). These algorithms can make use of complex features and factor graphs and yet benefit from favorable learning guarantees. We also report the results of experiments with VCRF on several datasets to validate our theory.

## 1 Introduction

Structured prediction covers a broad family of important learning problems. These include key tasks in natural language processing such as part-of-speech tagging, parsing, machine translation, and named-entity recognition, important areas in computer vision such as image segmentation and object recognition, and also crucial areas in speech processing such as pronunciation modeling and speech recognition.

In all these problems, the output space admits some structure. This may be a sequence of tags as in part-of-speech tagging, a parse tree as in context-free parsing, an acyclic graph as in dependency parsing, or labels of image segments as in object detection. Another property common to these tasks is that, in each case, the natural loss function admits a decomposition along the output substructures. As an example, the loss function may be the Hamming loss as in part-of-speech tagging, or it may be the edit-distance, which is widely used in natural language and speech processing.

The output structure and corresponding loss function make these problems significantly different from the (unstructured) binary classification problems extensively studied in learning theory. In recent years, a number of different algorithms have been designed for structured prediction, including Conditional Random Field (CRF) [Lafferty et al., 2001], StructSVM [Tsochantaridis et al., 2005], Maximum-Margin Markov Network (M3N) [Taskar et al., 2003], a kernel-regression algorithm [Cortes et al., 2007], and search-based approaches such as [Daumé III et al., 2009, Doppa et al., 2014, Lam et al., 2015, Chang et al., 2015, Ross et al., 2011]. More recently, deep learning techniques have also been developed for tasks including part-of-speech tagging [Jurafsky and Martin, 2009, Vinyals et al., 2015a], named-entity recognition [Nadeau and Sekine, 2007], machine translation [Zhang et al., 2008], image segmentation [Lucchi et al., 2013], and image annotation [Vinyals et al., 2015b].

However, in contrast to the plethora of algorithms, there have been relatively few studies devoted to the theoretical understanding of structured prediction [Bakir et al., 2007]. Existing learning guarantees hold primarily for simple losses such as the Hamming loss [Taskar et al., 2003, Cortes et al., 2014, Collins, 2001] and do not cover other natural losses such as the edit-distance. They also typically only apply to specific factor graph models. The main exception is the work of McAllester [2007], which provides PAC-Bayesian guarantees for arbitrary losses, though only in the special case of randomized algorithms using linear (count-based) hypotheses.

This paper presents a general theoretical analysis of structured prediction with a series of new results. We give new data-dependent margin guarantees for structured prediction for a broad family of loss functions and a general family of hypotheses, with an arbitrary factor graph decomposition. These are the tightest margin bounds known for both standard multi-class and general structured prediction problems. For special cases studied in the past, our learning bounds match or improve upon the previously best bounds (see Section 3.3). In particular, our bounds improve upon those of Taskar et al. [2003]. Our guarantees are expressed in terms of a data-dependent complexity measure, *factor graph complexity*, which we show can be estimated from data and bounded in terms of familiar quantities for several commonly used hypothesis sets along with a sparsity measure for features and graphs.

We further extend our theory by leveraging the principle of Voted Risk Minimization (VRM) and show that learning is possible even with complex factor graphs. We present new learning bounds for this advanced setting, which we use to design two new algorithms, *Voted Conditional Random Field* (VCRF) and *Voted Structured Boosting* (StructBoost). These algorithms can make use of complex features and factor graphs and yet benefit from favorable learning guarantees. As a proof of concept validating our theory, we also report the results of experiments with VCRF on several datasets.

The paper is organized as follows. In Section 2 we introduce the notation and definitions relevant to our discussion of structured prediction. In Section 3, we derive a series of new learning guarantees for structured prediction, which are then used to prove the VRM principle in Section 4. Section 5 develops the algorithmic framework which is directly based on our theory. In Section 6, we provide some preliminary experimental results that serve as a proof of concept for our theory.

## 2 Preliminaries

Let $\mathcal{X}$ denote the input space and $\mathcal{Y}$ the output space. In structured prediction, the output space may be a set of sequences, images, graphs, parse trees, lists, or some other (typically discrete) objects admitting some possibly overlapping structure. Thus, we assume that the output structure can be decomposed into $l$ substructures. For example, this may be positions along a sequence, so that the output space $\mathcal{Y}$ is decomposable along these substructures: $\mathcal{Y} = \mathcal{Y}_1 \times \cdots \times \mathcal{Y}_l$. Here, $\mathcal{Y}_k$ is the set of possible labels (or classes) that can be assigned to substructure $k$.

**Loss functions**. We denote by $\mathsf{L} \colon \mathcal{Y} \times \mathcal{Y} \to \mathbb{R}_+$ a loss function measuring the dissimilarity of two elements of the output space $\mathcal{Y}$. We will assume that the loss function $\mathsf{L}$ is *definite*, that is $\mathsf{L}(y, y') = 0$ iff $y = y'$. This assumption holds for all loss functions commonly used in structured prediction. A key aspect of structured prediction is that the loss function can be decomposed along the substructures $\mathcal{Y}_k$. As an example, $\mathsf{L}$ may be the Hamming loss defined by $\mathsf{L}(y, y') = \frac{1}{l} \sum_{k=1}^{l} 1_{y_k \neq y'_k}$ for all $y = (y_1, \ldots, y_l)$ and $y' = (y'_1, \ldots, y'_l)$, with $y_k, y'_k \in \mathcal{Y}_k$. In the common case where $\mathcal{Y}$ is a set of sequences defined over a finite alphabet, $\mathsf{L}$ may be the edit-distance, which is widely used in natural language and speech processing applications, with possibly different costs associated to insertions, deletions and substitutions. $\mathsf{L}$ may also be a loss based on the negative inner product of the vectors of $n$-gram counts of two sequences, or its negative logarithm. Such losses have been

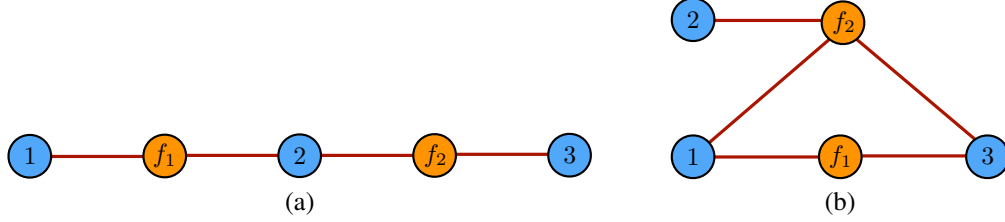

Figure 1: Example of factor graphs. (a) Pairwise Markov network decomposition: $h(x, y) = h_{f_1}(x, y_1, y_2) + h_{f_2}(x, y_2, y_3)$ (b) Other decomposition $h(x, y) = h_{f_1}(x, y_1, y_3) + h_{f_2}(x, y_1, y_2, y_3)$.

used to approximate the BLEU score loss in machine translation. There are other losses defined in computational biology based on various string-similarity measures. Our theoretical analysis is general and applies to arbitrary bounded and definite loss functions.

**Scoring functions and factor graphs**. We will adopt the common approach in structured prediction where predictions are based on a *scoring function* mapping $\mathcal{X} \times \mathcal{Y}$ to $\mathbb{R}$. Let $\mathcal{H}$ be a family of scoring functions. For any $h \in \mathcal{H}$, we denote by h the predictor defined by $h$: for any $x \in \mathcal{X}$, $\mathsf{h}(x) = \operatorname{argmax}_{y \in \mathcal{Y}} h(x, y)$.

Furthermore, we will assume, as is standard in structured prediction, that each function $h \in \mathcal{H}$ can be decomposed as a sum. We will consider the most general case for such decompositions, which can be made explicit using the notion of *factor graphs*.[1] A factor graph $G$ is a tuple $G = (V, F, E)$, where $V$ is a set of variable nodes, $F$ a set of factor nodes, and $E$ a set of undirected edges between a variable node and a factor node. In our context, $V$ can be identified with the set of substructure indices, that is $V = \{1, \ldots, l\}$.

For any factor node $f$, denote by $\mathcal{N}(f) \subseteq V$ the set of variable nodes connected to $f$ via an edge and define $\mathcal{Y}_f$ as the substructure set cross-product $\mathcal{Y}_f = \prod_{k \in \mathcal{N}(f)} \mathcal{Y}_k$. Then, $h$ admits the following decomposition as a sum of functions $h_f$, each taking as argument an element of the input space $x \in \mathcal{X}$ and an element of $\mathcal{Y}_f$, $y_f \in \mathcal{Y}_f$:

$$h(x, y) = \sum_{f \in F} h_f(x, y_f). \tag{1}$$

Figure 1 illustrates this definition with two different decompositions. More generally, we will consider the setting in which a factor graph may depend on a particular example $(x_i, y_i)$: $G(x_i, y_i) = G_i = ([l_i], F_i, E_i)$. A special case of this setting is for example when the size $l_i$ (or length) of each example is allowed to vary and where the number of possible labels $|\mathcal{Y}|$ is potentially infinite.

We present other examples of such hypothesis sets and their decomposition in Section 3, where we discuss our learning guarantees. Note that such hypothesis sets $\mathcal{H}$ with an additive decomposition are those commonly used in most structured prediction algorithms [Tsochantaridis et al., 2005, Taskar et al., 2003, Lafferty et al., 2001]. This is largely motivated by the computational requirement for efficient training and inference. Our results, while very general, further provide a statistical learning motivation for such decompositions.

**Learning scenario**. We consider the familiar supervised learning scenario where the training and test points are drawn i.i.d. according to some distribution $\mathcal{D}$ over $\mathcal{X} \times \mathcal{Y}$. We will further adopt the standard definitions of margin, generalization error and empirical error. The margin $\rho_h(x, y)$ of a hypothesis $h$ for a labeled example $(x, y) \in \mathcal{X} \times \mathcal{Y}$ is defined by

$$\rho_h(x, y) = h(x, y) - \max_{y' \neq y} h(x, y'). \tag{2}$$

Let $S = ((x_1, y_1), \ldots, (x_m, y_m))$ be a training sample of size $m$ drawn from $\mathcal{D}^m$. We denote by $R(h)$ the generalization error and by $\widehat{R}_S(h)$ the empirical error of $h$ over $S$:

$$R(h) = \mathop{\mathbb{E}}_{(x,y) \sim \mathcal{D}} [\mathsf{L}(\mathsf{h}(x), y)] \quad \text{and} \quad \widehat{R}_S(h) = \mathop{\mathbb{E}}_{(x,y) \sim S} [\mathsf{L}(\mathsf{h}(x), y)], \tag{3}$$

where $\mathsf{h}(x) = \operatorname{argmax}_y h(x, y)$ and where the notation $(x, y) \sim S$ indicates that $(x, y)$ is drawn according to the empirical distribution defined by $S$. The learning problem consists of using the sample $S$ to select a hypothesis $h \in \mathcal{H}$ with small expected loss $R(h)$.

Observe that the definiteness of the loss function implies, for all $x \in \mathcal{X}$, the following equality:

$$\mathsf{L}(\mathsf{h}(x), y) = \mathsf{L}(\mathsf{h}(x), y)\, 1_{\rho_h(x,y) \leq 0}. \tag{4}$$

We will later use this identity in the derivation of surrogate loss functions.

## 3 General learning bounds for structured prediction

In this section, we present new learning guarantees for structured prediction. Our analysis is general and applies to the broad family of definite and bounded loss functions described in the previous section. It is also general in the sense that it applies to general hypothesis sets and not just sub-families of linear functions. For linear hypotheses, we will give a more refined analysis that holds for arbitrary norm-$p$ regularized hypothesis sets.

The theoretical analysis of structured prediction is more complex than for classification since, by definition, it depends on the properties of the loss function and the factor graph. These attributes capture the combinatorial properties of the problem which must be exploited since the total number of labels is often exponential in the size of that graph. To tackle this problem, we first introduce a new complexity tool.

### 3.1 Complexity measure

A key ingredient of our analysis is a new data-dependent notion of complexity that extends the classical Rademacher complexity. We define the *empirical factor graph Rademacher complexity* $\widehat{\mathfrak{R}}_S^G(\mathcal{H})$ of a hypothesis set $\mathcal{H}$ for a sample $S = (x_1, \ldots, x_m)$ and factor graph $G$ as follows:

$$\widehat{\mathfrak{R}}_S^G(\mathcal{H}) = \frac{1}{m} \mathop{\mathbb{E}}_{\boldsymbol{\epsilon}} \left[ \sup_{h \in \mathcal{H}} \sum_{i=1}^m \sum_{f \in F_i} \sum_{y \in \mathcal{Y}_f} \sqrt{|F_i|}\, \epsilon_{i,f,y}\, h_f(x_i, y) \right],$$

where $\boldsymbol{\epsilon} = (\epsilon_{i,f,y})_{i \in [m], f \in F_i, y \in \mathcal{Y}_f}$ and where $\epsilon_{i,f,y}$s are independent Rademacher random variables uniformly distributed over $\{\pm 1\}$. The *factor graph Rademacher complexity* of $\mathcal{H}$ for a factor graph $G$ is defined as the expectation: $\mathfrak{R}_m^G(\mathcal{H}) = \mathbb{E}_{S \sim \mathcal{D}^m}\big[\widehat{\mathfrak{R}}_S^G(\mathcal{H})\big]$. It can be shown that the empirical factor graph Rademacher complexity is concentrated around its mean (Lemma 8). The factor graph Rademacher complexity is a natural extension of the standard Rademacher complexity to vector-valued hypothesis sets (with one coordinate per factor in our case). For binary classification, the factor graph and standard Rademacher complexities coincide. Otherwise, the factor graph complexity can be upper bounded in terms of the standard one. As with the standard Rademacher complexity, the factor graph Rademacher complexity of a hypothesis set can be estimated from data in many cases. In some important cases, it also admits explicit upper bounds similar to those for the standard Rademacher complexity but with an additional dependence on the factor graph quantities. We will prove this for several families of functions which are commonly used in structured prediction (Theorem 2).

### 3.2 Generalization bounds

In this section, we present new margin bounds for structured prediction based on the factor graph Rademacher complexity of $\mathcal{H}$. Our results hold both for the additive and the multiplicative empirical margin losses defined below:

$$\widehat{R}_{S,\rho}^{\mathrm{add}}(h) = \mathop{\mathbb{E}}_{(x,y) \sim S} \left[ \Phi^* \left( \max_{y' \neq y} \mathsf{L}(y', y) - \frac{1}{\rho}\big[h(x, y) - h(x, y')\big] \right) \right] \tag{5}$$

$$\widehat{R}_{S,\rho}^{\mathrm{mult}}(h) = \mathop{\mathbb{E}}_{(x,y) \sim S} \left[ \Phi^* \left( \max_{y' \neq y} \mathsf{L}(y', y) \Big( 1 - \frac{1}{\rho}[h(x, y) - h(x, y')] \Big) \right) \right]. \tag{6}$$

Here, $\Phi^*(r) = \min(M, \max(0, r))$ for all $r$, with $M = \max_{y,y'} \mathsf{L}(y, y')$. As we show in Section 5, convex upper bounds on $\widehat{R}_{S,\rho}^{\mathrm{add}}(h)$ and $\widehat{R}_{S,\rho}^{\mathrm{mult}}(h)$ directly lead to many existing structured prediction algorithms. The following is our general data-dependent margin bound for structured prediction.

**Theorem 1.** *Fix $\rho > 0$. For any $\delta > 0$, with probability at least $1 - \delta$ over the draw of a sample $S$ of size $m$, the following holds for all $h \in \mathcal{H}$,*

$$R(h) \leq R_\rho^{add}(h) \leq \widehat{R}_{S,\rho}^{add}(h) + \frac{4\sqrt{2}}{\rho}\mathfrak{R}_m^G(\mathcal{H}) + M\sqrt{\frac{\log \frac{1}{\delta}}{2m}},$$

$$R(h) \leq R_\rho^{mult}(h) \leq \widehat{R}_{S,\rho}^{mult}(h) + \frac{4\sqrt{2}M}{\rho}\mathfrak{R}_m^G(\mathcal{H}) + M\sqrt{\frac{\log \frac{1}{\delta}}{2m}}.$$

The full proof of Theorem 1 is given in Appendix A. It is based on a new contraction lemma (Lemma 5) generalizing Talagrand's lemma that can be of independent interest.[2] We also present a more refined contraction lemma (Lemma 6) that can be used to improve the bounds of Theorem 1. Theorem 1 is the first data-dependent generalization guarantee for structured prediction with general loss functions, general hypothesis sets, and arbitrary factor graphs for both multiplicative and additive margins. We also present a version of this result with empirical complexities as Theorem 7 in the supplementary material. We will compare these guarantees to known special cases below.

The margin bounds above can be extended to hold uniformly over $\rho \in (0, 1]$ at the price of an additional term of the form $\sqrt{(\log \log_2 \frac{2}{\rho})/m}$ in the bound, using known techniques (see for example [Mohri et al., 2012]).

The hypothesis set used by convex structured prediction algorithms such as StructSVM [Tsochantaridis et al., 2005], Max-Margin Markov Networks (M3N) [Taskar et al., 2003] or Conditional Random Field (CRF) [Lafferty et al., 2001] is that of linear functions. More precisely, let $\mathbf{\Psi}$ be a feature mapping from $(\mathcal{X} \times \mathcal{Y})$ to $\mathbb{R}^N$ such that $\mathbf{\Psi}(x, y) = \sum_{f \in F} \mathbf{\Psi}_f(x, y_f)$. For any $p$, define $\mathcal{H}_p$ as follows:

$$\mathcal{H}_p = \{x \mapsto \mathbf{w} \cdot \mathbf{\Psi}(x, y) \colon \mathbf{w} \in \mathbb{R}^N, \|\mathbf{w}\|_p \leq \Lambda_p\}.$$

Then, $\widehat{\mathfrak{R}}_m^G(\mathcal{H}_p)$ can be efficiently estimated using random sampling and solving LP programs. Moreover, one can obtain explicit upper bounds on $\widehat{\mathfrak{R}}_m^G(\mathcal{H}_p)$. To simplify our presentation, we will consider the case $p = 1, 2$, but our results can be extended to arbitrary $p \geq 1$ and, more generally, to arbitrary group norms.

**Theorem 2.** *For any sample $S = (x_1, \ldots, x_m)$, the following upper bounds hold for the empirical factor graph complexity of $\mathcal{H}_1$ and $\mathcal{H}_2$:*

$$\widehat{\mathfrak{R}}_S^G(\mathcal{H}_1) \leq \frac{\Lambda_1 r_\infty}{m}\sqrt{s \log(2N)}, \qquad \widehat{\mathfrak{R}}_S^G(\mathcal{H}_2) \leq \frac{\Lambda_2 r_2}{m}\sqrt{\textstyle\sum_{i=1}^m \sum_{f \in F_i} \sum_{y \in \mathcal{Y}_f} |F_i|},$$

*where $r_\infty = \max_{i,f,y} \|\Psi_f(x_i, y)\|_\infty$, $r_2 = \max_{i,f,y} \|\Psi_f(x_i, y)\|_2$ and where $s$ is a sparsity factor defined by $s = \max_{j \in [1,N]} \sum_{i=1}^m \sum_{f \in F_i} \sum_{y \in \mathcal{Y}_f} |F_i| 1_{\Psi_{f,j}(x_i,y) \neq 0}$.*

Plugging in these factor graph complexity upper bounds into Theorem 1 immediately yields explicit data-dependent structured prediction learning guarantees for linear hypotheses with general loss functions and arbitrary factor graphs (see Corollary 10). Observe that, in the worst case, the sparsity factor can be bounded as follows:

$$s \leq \sum_{i=1}^m \sum_{f \in F_i} \sum_{y \in \mathcal{Y}_f} |F_i| \leq \sum_{i=1}^m |F_i|^2 d_i \leq m \max_i |F_i|^2 d_i,$$

where $d_i = \max_{f \in F_i} |\mathcal{Y}_f|$. Thus, the factor graph Rademacher complexities of linear hypotheses in $\mathcal{H}_1$ scale as $O(\sqrt{\log(N) \max_i |F_i|^2 d_i/m})$. An important observation is that $|F_i|$ and $d_i$ depend on the observed sample. This shows that the *expected size* of the factor graph is crucial for learning in this scenario. This should be contrasted with other existing structured prediction guarantees that we discuss below, which assume a fixed upper bound on the size of the factor graph. Note that our result shows that learning is possible even with an infinite set $\mathcal{Y}$. To the best of our knowledge, this is the first learning guarantee for learning with infinitely many classes.

Our learning guarantee for $\mathcal{H}_1$ can additionally benefit from the sparsity of the feature mapping and observed data. In particular, in many applications, $\Psi_{f,j}$ is a binary indicator function that is non-zero for a single $(x,y) \in \mathcal{X} \times \mathcal{Y}_f$. For instance, in NLP, $\Psi_{f,j}$ may indicate an occurrence of a certain $n$-gram in the input $x_i$ and output $y_i$. In this case, $s = \sum_{i=1}^m |F_i|^2 \leq m \max_i |F_i|^2$ and the complexity term is only in $O(\max_i |F_i| \sqrt{\log(N)/m})$, where $N$ may depend linearly on $d_i$.

## 3.3 Special cases and comparisons

**Markov networks**. For the pairwise Markov networks with a fixed number of substructures $l$ studied by Taskar et al. [2003], our equivalent factor graph admits $l$ nodes, $|F_i| = l$, and the maximum size of $\mathcal{Y}_f$ is $d_i = k^2$ if each substructure of a pair can be assigned one of $k$ classes. Thus, if we apply Corollary 10 with Hamming distance as our loss function and divide the bound through by $l$, to normalize the loss to interval $[0,1]$ as in [Taskar et al., 2003], we obtain the following explicit form of our guarantee for an additive empirical margin loss, for all $h \in \mathcal{H}_2$:

$$R(h) \leq \widehat{R}_{S,\rho}^{\text{add}}(h) + \frac{4\Lambda_2 r_2}{\rho} \sqrt{\frac{2k^2}{m}} + 3\sqrt{\frac{\log \frac{1}{\delta}}{2m}}.$$

This bound can be further improved by eliminating the dependency on $k$ using an extension of our contraction Lemma 5 to $\| \cdot \|_{\infty,2}$ (see Lemma 6). The complexity term of Taskar et al. [2003] is bounded by a quantity that varies as $\widetilde{O}(\sqrt{\Lambda_2^2 q^2 r_2^2 / m})$, where $q$ is the maximal out-degree of a factor graph. Our bound has the same dependence on these key quantities, but with no logarithmic term in our case. Note that, unlike the result of Taskar et al. [2003], our bound also holds for general loss functions and different $p$-norm regularizers. Moreover, our result for a multiplicative empirical margin loss is new, even in this special case.

**Multi-class classification**. For standard (unstructured) multi-class classification, we have $|F_i| = 1$ and $d_i = c$, where $c$ is the number of classes. In that case, for linear hypotheses with norm-2 regularization, the complexity term of our bound varies as $O(\Lambda_2 r_2 \sqrt{c/\rho^2 m})$ (Corollary 11). This improves upon the best known general margin bounds of Kuznetsov et al. [2014], who provide a guarantee that scales linearly with the number of classes instead. Moreover, in the special case where an individual $\mathbf{w}_y$ is learned for each class $y \in [c]$, we retrieve the recent favorable bounds given by Lei et al. [2015], albeit with a somewhat simpler formulation. In that case, for any $(x,y)$, all components of the feature vector $\Psi(x,y)$ are zero, except (perhaps) for the $N$ components corresponding to class $y$, where $N$ is the dimension of $\mathbf{w}_y$. In view of that, for example for a group-norm $\| \cdot \|_{2,1}$-regularization, the complexity term of our bound varies as $O(\Lambda r \sqrt{(\log c)/\rho^2 m})$, which matches the results of Lei et al. [2015] with a logarithmic dependency on $c$ (ignoring some complex exponents of $\log c$ in their case). Additionally, note that unlike existing multi-class learning guarantees, our results hold for arbitrary loss functions. See Corollary 12 for further details. Our sparsity-based bounds can also be used to give bounds with logarithmic dependence on the number of classes when the features only take values in $\{0,1\}$. Finally, using Lemma 6 instead of Lemma 5, the dependency on the number of classes can be further improved.

We conclude this section by observing that, since our guarantees are expressed in terms of the average size of the factor graph over a given sample, this invites us to search for a hypothesis set $\mathcal{H}$ and predictor $h \in \mathcal{H}$ such that the tradeoff between the empirical size of the factor graph and empirical error is optimal. In the next section, we will make use of the recently developed principle of Voted Risk Minimization (VRM) [Cortes et al., 2015] to reach this objective.

# 4 Voted Risk Minimization

In many structured prediction applications such as natural language processing and computer vision, one may wish to exploit very rich features. However, the use of rich families of hypotheses could lead to overfitting. In this section, we show that it may be possible to use rich families in conjunction with simpler families, provided that *fewer* complex hypotheses are used (or that they are used with less mixture weight). We achieve this goal by deriving learning guarantees for ensembles of structured prediction rules that explicitly account for the differing complexities between families. This will motivate the algorithms that we present in Section 5.

Assume that we are given $p$ families $H_1, \ldots, H_p$ of functions mapping from $\mathcal{X} \times \mathcal{Y}$ to $\mathbb{R}$. Define the ensemble family $\mathcal{F} = \text{conv}(\cup_{k=1}^{p} H_k)$, that is the family of functions $f$ of the form $f = \sum_{t=1}^{T} \alpha_t h_t$, where $\boldsymbol{\alpha} = (\alpha_1, \ldots, \alpha_T)$ is in the simplex $\Delta$ and where, for each $t \in [1, T]$, $h_t$ is in $H_{k_t}$ for some $k_t \in [1, p]$. We further assume that $\mathfrak{R}_m^G(H_1) \leq \mathfrak{R}_m^G(H_2) \leq \ldots \leq \mathfrak{R}_m^G(H_p)$. As an example, the $H_k$s may be ordered by the size of the corresponding factor graphs.

The main result of this section is a generalization of the VRM theory to the structured prediction setting. The learning guarantees that we present are in terms of upper bounds on $\widehat{R}_{S,\rho}^{\text{add}}(h)$ and $\widehat{R}_{S,\rho}^{\text{mult}}(h)$, which are defined as follows for all $\tau \geq 0$:

$$\widehat{R}_{S,\rho,\tau}^{\text{add}}(h) = \mathop{\mathbb{E}}_{(x,y)\sim S}\left[\Phi^*\left(\max_{y'\neq y} \mathsf{L}(y', y) + \tau - \frac{1}{\rho}\big[h(x,y) - h(x,y')\big]\right)\right] \quad (7)$$

$$\widehat{R}_{S,\rho,\tau}^{\text{mult}}(h) = \mathop{\mathbb{E}}_{(x,y)\sim S}\left[\Phi^*\left(\max_{y'\neq y} \mathsf{L}(y', y)\Big(1 + \tau - \frac{1}{\rho}[h(x,y) - h(x,y')]\Big)\right)\right]. \quad (8)$$

Here, $\tau$ can be interpreted as a margin term that acts in conjunction with $\rho$. For simplicity, we assume in this section that $|\mathcal{Y}| = c < +\infty$.

**Theorem 3.** *Fix $\rho > 0$. For any $\delta > 0$, with probability at least $1 - \delta$ over the draw of a sample $S$ of size $m$, each of the following inequalities holds for all $f \in \mathcal{F}$:*

$$R(f) - \widehat{R}_{S,\rho,1}^{add}(f) \leq \frac{4\sqrt{2}}{\rho}\sum_{t=1}^{T}\alpha_t\mathfrak{R}_m^G(H_{k_t}) + C(\rho, M, c, m, p),$$

$$R(f) - \widehat{R}_{S,\rho,1}^{mult}(f) \leq \frac{4\sqrt{2}M}{\rho}\sum_{t=1}^{T}\alpha_t\mathfrak{R}_m^G(H_{k_t}) + C(\rho, M, c, m, p),$$

*where $C(\rho, M, c, m, p) = \frac{2M}{\rho}\sqrt{\frac{\log p}{m}} + 3M\sqrt{\left\lceil\frac{4}{\rho^2}\log\left(\frac{c^2\rho^2 m}{4\log p}\right)\right\rceil\frac{\log p}{m} + \frac{\log\frac{2}{\delta}}{2m}}$.*

The proof of this theorem crucially depends on the theory we developed in Section 3 and is given in Appendix A. As with Theorem 1, we also present a version of this result with empirical complexities as Theorem 14 in the supplementary material. The explicit dependence of this bound on the parameter vector $\boldsymbol{\alpha}$ suggests that learning even with highly complex hypothesis sets could be possible so long as the complexity term, which is a weighted average of the factor graph complexities, is not too large. The theorem provides a quantitative way of determining the mixture weights that should be apportioned to each family. Furthermore, the dependency on the number of distinct feature map families $H_k$ is very mild and therefore suggests that a large number of families can be used. These properties will be useful for motivating new algorithms for structured prediction.

## 5 Algorithms

In this section, we derive several algorithms for structured prediction based on the VRM principle discussed in Section 4. We first give general convex upper bounds (Section 5.1) on the structured prediction loss which recover as special cases the loss functions used in StructSVM [Tsochantaridis et al., 2005], Max-Margin Markov Networks (M3N) [Taskar et al., 2003], and Conditional Random Field (CRF) [Lafferty et al., 2001]. Next, we introduce a new algorithm, Voted Conditional Random Field (VCRF) Section 5.2, with accompanying experiments as proof of concept. We also present another algorithm, Voted StructBoost (VStructBoost), in Appendix C.

### 5.1 General framework for convex surrogate losses

Given $(x, y) \in \mathcal{X} \times \mathcal{Y}$, the mapping $h \mapsto \mathsf{L}(\mathsf{h}(x), y)$ is typically not a convex function of $h$, which leads to computationally hard optimization problems. This motivates the use of convex surrogate losses. We first introduce a general formulation of surrogate losses for structured prediction problems.

**Lemma 4.** *For any $u \in \mathbb{R}_+$, let $\Phi_u \colon \mathbb{R} \to \mathbb{R}$ be an upper bound on $v \mapsto u\mathbf{1}_{v\leq 0}$. Then, the following upper bound holds for any $h \in \mathcal{H}$ and $(x, y) \in \mathcal{X} \times \mathcal{Y}$,*

$$\mathsf{L}(\mathsf{h}(x), y) \leq \max_{y'\neq y} \Phi_{\mathsf{L}(y',y)}(h(x,y) - h(x,y')). \quad (9)$$

The proof is given in Appendix A. This result defines a general framework that enables us to straightforwardly recover many of the most common state-of-the-art structured prediction algorithms via suitable choices of $\Phi_u(v)$: (a) for $\Phi_u(v) = \max(0, u(1-v))$, the right-hand side of (9) coincides with the surrogate loss defining StructSVM [Tsochantaridis et al., 2005]; (b) for $\Phi_u(v) = \max(0, u - v)$, it coincides with the surrogate loss defining Max-Margin Markov Networks (M3N) [Taskar et al., 2003] when using for L the Hamming loss; and (c) for $\Phi_u(v) = \log(1 + e^{u-v})$, it coincides with the surrogate loss defining the Conditional Random Field (CRF) [Lafferty et al., 2001].

Moreover, alternative choices of $\Phi_u(v)$ can help define new algorithms. In particular, we will refer to the algorithm based on the surrogate loss defined by $\Phi_u(v) = ue^{-v}$ as *StructBoost*, in reference to the exponential loss used in AdaBoost. Another related alternative is based on the choice $\Phi_u(v) = e^{u-v}$. See Appendix C, for further details on this algorithm. In fact, for each $\Phi_u(v)$ described above, the corresponding convex surrogate is an upper bound on either the multiplicative or additive margin loss introduced in Section 3. Therefore, each of these algorithms seeks a hypothesis that minimizes the generalization bounds presented in Section 3. To the best of our knowledge, this interpretation of these well-known structured prediction algorithms is also new. In what follows, we derive new structured prediction algorithms that minimize finer generalization bounds presented in Section 4.

## 5.2 Voted Conditional Random Field (VCRF)

We first consider the convex surrogate loss based on $\Phi_u(v) = \log(1 + e^{u-v})$, which corresponds to the loss defining CRF models. Using the monotonicity of the logarithm and upper bounding the maximum by a sum gives the following upper bound on the surrogate loss holds:

$$\max_{y' \neq y} \log(1 + e^{\mathsf{L}(y,y') - \mathbf{w} \cdot (\mathbf{\Psi}(x,y) - \mathbf{\Psi}(x,y'))}) \leq \log \bigg( \sum_{y' \in \mathcal{Y}} e^{\mathsf{L}(y,y') - \mathbf{w} \cdot (\mathbf{\Psi}(x,y) - \mathbf{\Psi}(x,y'))} \bigg),$$

which, combined with VRM principle leads to the following optimization problem:

$$\min_{\mathbf{w}} \frac{1}{m} \sum_{i=1}^{m} \log \bigg( \sum_{y \in \mathcal{Y}} e^{\mathsf{L}(y,y_i) - \mathbf{w} \cdot (\mathbf{\Psi}(x_i,y_i) - \mathbf{\Psi}(x_i,y))} \bigg) + \sum_{k=1}^{p} (\lambda r_k + \beta) \|\mathbf{w}_k\|_1, \qquad (10)$$

where $r_k = r_\infty |F(k)| \sqrt{\log N}$. We refer to the learning algorithm based on the optimization problem (10) as VCRF. Note that for $\lambda = 0$, (10) coincides with the objective function of $L_1$-regularized CRF. Observe that we can also directly use $\max_{y' \neq y} \log(1 + e^{\mathsf{L}(y,y') - \mathbf{w} \cdot \delta\mathbf{\Psi}(x,y,y')})$ or its upper bound $\sum_{y' \neq y} \log(1 + e^{\mathsf{L}(y,y') - \mathbf{w} \cdot \delta\mathbf{\Psi}(x,y,y')})$ as a convex surrogate. We can similarly derive an $L_2$-regularization formulation of the VCRF algorithm. In Appendix D, we describe efficient algorithms for solving the VCRF and VStructBoost optimization problems.

# 6 Experiments

In Appendix B, we corroborate our theory by reporting experimental results suggesting that the VCRF algorithm can outperform the CRF algorithm on a number of part-of-speech (POS) datasets.

# 7 Conclusion

We presented a general theoretical analysis of structured prediction. Our data-dependent margin guarantees for structured prediction can be used to guide the design of new algorithms or to derive guarantees for existing ones. Its explicit dependency on the properties of the factor graph and on feature sparsity can help shed new light on the role played by the graph and features in generalization. Our extension of the VRM theory to structured prediction provides a new analysis of generalization when using a very rich set of features, which is common in applications such as natural language processing and leads to new algorithms, VCRF and VStructBoost. Our experimental results for VCRF serve as a proof of concept and motivate more extensive empirical studies of these algorithms.

## Acknowledgments

This work was partly funded by NSF CCF-1535987 and IIS-1618662 and NSF GRFP DGE-1342536.

## Footnotes

[1]Factor graphs are typically used to indicate the factorization of a probabilistic model. We are not assuming probabilistic models, but they would be also captured by our general framework: $h$ would then be - log of a probability.

[2]A result similar to Lemma 5 has also been recently proven independently in [Maurer, 2016].

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
