[Supplementary Material]

# A  Proofs

This appendix section gathers detailed proofs of all of our main results. In Appendix A.1, we prove a contraction lemma used as a tool in the proof of our general factor graph Rademacher complexity bounds (Appendix A.3). In Appendix A.8, we further extend our bounds to the Voted Risk Minimization setting. Appendix A.5 gives explicit upper bounds on the factor graph Rademacher complexity of several commonly used hypothesis sets. In Appendix A.9, we prove a general upper bound on a loss function used in structured prediction in terms of a convex surrogate.

## A.1  Contraction lemma

The following contraction lemma will be a key tool used in the proofs of our generalization bounds for structured prediction.

**Lemma 5.** *Let $\mathcal{H}$ be a hypothesis set of functions mapping $\mathcal{X}$ to $\mathbb{R}^c$. Assume that for all $i = 1, \ldots, m$, $\Psi_i : \mathbb{R}^c \to \mathbb{R}$ is $\mu_i$-Lipschitz for $\mathbb{R}^c$ equipped with the 2-norm. That is:*

$$|\Psi_i(\mathbf{x}') - \Psi_i(\mathbf{x})| \le \mu_i \|\mathbf{x}' - \mathbf{x}\|_2,$$

*for all $(\mathbf{x}, \mathbf{x}') \in (\mathbb{R}^c)^2$. Then, for any sample $S$ of $m$ points $x_1, \ldots, x_m \in \mathcal{X}$, the following inequality holds*

$$\frac{1}{m} \mathop{\mathbb{E}}_{\boldsymbol{\sigma}} \left[ \sup_{\mathbf{h} \in \mathcal{H}} \sum_{i=1}^m \sigma_i \Psi_i(\mathbf{h}(x_i)) \right] \le \frac{\sqrt{2}}{m} \mathop{\mathbb{E}}_{\boldsymbol{\epsilon}} \left[ \sup_{\mathbf{h} \in \mathcal{H}} \sum_{i=1}^m \sum_{j=1}^c \epsilon_{ij} \, \mu_i h_j(x_i) \right], \tag{11}$$

*where $\boldsymbol{\epsilon} = (\epsilon_{ij})_{i,j}$ and $\epsilon_{ij}$s are independent Rademacher variables uniformly distributed over $\{\pm 1\}$.*

*Proof.* Fix a sample $S = (x_1, \ldots, x_m)$. Then, we can rewrite the left-hand side of (11) as follows:

$$\frac{1}{m} \mathop{\mathbb{E}}_{\boldsymbol{\sigma}} \left[ \sup_{\mathbf{h} \in \mathcal{H}} \sum_{i=1}^m \sigma_i \Psi_i(\mathbf{h}(x_i)) \right] = \frac{1}{m} \mathop{\mathbb{E}}_{\sigma_1, \ldots, \sigma_{m-1}} \left[ \mathop{\mathbb{E}}_{\sigma_m} \left[ \sup_{\mathbf{h} \in \mathcal{H}} U_{m-1}(\mathbf{h}) + \sigma_m \Psi_m(\mathbf{h}(x_m)) \right] \right],$$

where $U_{m-1}(\mathbf{h}) = \sum_{i=1}^{m-1} \sigma_i \Psi_i(\mathbf{h}(x_i))$. Assume that the suprema can be attained and let $\mathbf{h}_1, \mathbf{h}_2 \in \mathcal{H}$ be the hypotheses satisfying

$$U_{m-1}(\mathbf{h}_1) + \Psi_m(\mathbf{h}_1(x_m)) = \sup_{\mathbf{h} \in \mathcal{H}} U_{m-1}(\mathbf{h}) + \Psi_m(\mathbf{h}(x_m))$$

$$U_{m-1}(\mathbf{h}_2) - \Psi_m(\mathbf{h}_2(x_m)) = \sup_{\mathbf{h} \in \mathcal{H}} U_{m-1}(\mathbf{h}) - \Psi_m(\mathbf{h}(x_m)).$$

When the suprema are not reached, a similar argument to what follows can be given by considering instead hypotheses that are $\epsilon$-close to the suprema for any $\epsilon > 0$. By definition of expectation, since $\sigma_m$ is uniformly distributed over $\{\pm 1\}$, we can write

$$\mathop{\mathbb{E}}_{\sigma_m} \left[ \sup_{\mathbf{h} \in \mathcal{H}} U_{m-1}(\mathbf{h}) + \sigma_m \Psi_m(\mathbf{h}(x_m)) \right]$$

$$= \frac{1}{2} \sup_{\mathbf{h} \in \mathcal{H}} U_{m-1}(\mathbf{h}) + \Psi_m(\mathbf{h}(x_m)) + \frac{1}{2} \sup_{\mathbf{h} \in \mathcal{H}} U_{m-1}(\mathbf{h}) - \Psi_m(\mathbf{h}(x_m))$$

$$= \frac{1}{2}[U_{m-1}(\mathbf{h}_1) + \Psi_m(\mathbf{h}_1(x_m))] + \frac{1}{2}[U_{m-1}(\mathbf{h}_2) - \Psi_m(\mathbf{h}_2(x_m))].$$

Next, using the $\mu_m$-Lipschitzness of $\Psi_m$ and the Khintchine-Kahane inequality, we can write

$$\mathop{\mathbb{E}}_{\sigma_m} \left[ \sup_{\mathbf{h} \in \mathcal{H}} U_{m-1}(\mathbf{h}) + \sigma_m \Psi_m(\mathbf{h}(x_m)) \right]$$

$$\le \frac{1}{2}[U_{m-1}(\mathbf{h}_1) + U_{m-1}(\mathbf{h}_2) + \mu_m \|\mathbf{h}_1(x_m) - \mathbf{h}_2(x_m)\|_2]$$

$$\le \frac{1}{2} \left[ U_{m-1}(\mathbf{h}_1) + U_{m-1}(\mathbf{h}_2) + \mu_m \sqrt{2} \mathop{\mathbb{E}}_{\epsilon_{m1}, \ldots, \epsilon_{mc}} \left[ \left| \sum_{j=1}^c \epsilon_{mj} \big( h_{1j}(x_m) - h_{2j}(x_m) \big) \right| \right] \right].$$

Now, let $\epsilon_m$ denote $(\epsilon_{m1}, \ldots, \epsilon_{mc})$ and let $s(\epsilon_m) \in \{\pm 1\}$ denote the sign of $\sum_{j=1}^c \epsilon_{mj}(h_{1j}(x_m) - h_{2j}(x_m))$. Then, the following holds:

$$
\mathop{\mathbb{E}}_{\sigma_m} \left[ \sup_{\mathbf{h} \in \mathcal{H}} U_{m-1}(\mathbf{h}) + \sigma_m (\Psi_m \circ h)(x_m) \right]
$$

$$
\leq \frac{1}{2} \mathop{\mathbb{E}}_{\epsilon_m} \left[ U_{m-1}(\mathbf{h}_1) + U_{m-1}(\mathbf{h}_2) + \mu_m \sqrt{2} \Big| \sum_{j=1}^c \epsilon_{mj}(h_{1j}(x_m) - h_{2j}(x_m)) \Big| \right]
$$

$$
= \frac{1}{2} \mathop{\mathbb{E}}_{\epsilon_m} \left[ U_{m-1}(\mathbf{h}_1) + \mu_m \sqrt{2}\, s(\epsilon_m) \sum_{j=1}^c \epsilon_{mj} h_{1j}(x_m) \right.
$$

$$
\left. + U_{m-1}(\mathbf{h}_2) - \mu_m \sqrt{2}\, s(\epsilon_m) \sum_{j=1}^c \epsilon_{mj} h_{2j}(x_m) \right]
$$

$$
\leq \frac{1}{2} \mathop{\mathbb{E}}_{\epsilon_m} \left[ \sup_{\mathbf{h} \in \mathcal{H}} \left( U_{m-1}(\mathbf{h}) + \mu_m \sqrt{2}\, s(\epsilon_m) \sum_{j=1}^c \epsilon_{mj} h_j(x_m) \right) \right.
$$

$$
\left. + \sup_{\mathbf{h} \in \mathcal{H}} \left( U_{m-1}(\mathbf{h}) - \mu_m \sqrt{2}\, s(\epsilon_m) \sum_{j=1}^c \epsilon_{mj} h_j(x_m) \right) \right]
$$

$$
= \mathop{\mathbb{E}}_{\epsilon_m} \left[ \mathop{\mathbb{E}}_{\sigma_m} \left[ \sup_{\mathbf{h} \in \mathcal{H}} U_{m-1}(\mathbf{h}) + \mu_m \sqrt{2}\, \sigma_m \sum_{j=1}^c \epsilon_{mj} h_j(x_m) \right] \right]
$$

$$
= \mathop{\mathbb{E}}_{\epsilon_m} \left[ \sup_{\mathbf{h} \in \mathcal{H}} U_{m-1}(\mathbf{h}) + \mu_m \sqrt{2} \sum_{j=1}^c \epsilon_{mj} h_j(x_m) \right],
$$

Proceeding in the same way for all other $\sigma_i$s ($i < m$) completes the proof. $\qquad\square$

## A.2 Contraction lemma for $\|\cdot\|_{\infty,2}$-norm

In this section, we present an extension of the contraction Lemma 5, that can be used to remove the dependency on the alphabet size in all of our bounds.

**Lemma 6.** *Let $\mathcal{H}$ be a hypothesis set of functions mapping $\mathcal{X} \times [d]$ to $\mathbb{R}^c$. Assume that for all $i = 1, \ldots, m$, $\Psi_i$ is $\mu_i$-Lipschitz for $\mathbb{R}^{c \times d}$ equipped with the norm-$(\infty, 2)$ for some $\mu_i > 0$. That is*

$$
|\Psi_i(\mathbf{x}') - \Psi_i(\mathbf{x})| \leq \mu_i \|\mathbf{x}' - \mathbf{x}\|_{\infty,2},
$$

*for all $(\mathbf{x}, \mathbf{x}') \in (\mathbb{R}^{c \times d})^2$. Then, for any sample $S$ of $m$ points $x_1, \ldots, x_m \in \mathcal{X}$, there exists a distribution $\mathcal{U}$ over $[d]^{c \times m}$ such that the following inequality holds:*

$$
\frac{1}{m} \mathop{\mathbb{E}}_{\boldsymbol{\sigma}} \left[ \sup_{\mathbf{h} \in \mathcal{H}} \sum_{i=1}^m \sigma_i \Psi_i(\mathbf{h}(x_i)) \right] \leq \frac{\sqrt{2}}{m} \mathop{\mathbb{E}}_{\boldsymbol{v} \sim \mathcal{U}, \boldsymbol{\epsilon}} \left[ \sup_{\mathbf{h} \in \mathcal{H}} \sum_{i=1}^m \sum_{j=1}^c \epsilon_{ij}\, \mu_i h_j(x_i, v_{mj}) \right], \qquad (12)
$$

*where $\boldsymbol{\epsilon} = (\epsilon_{ij})_{i,j}$ and $\epsilon_{ij}$s are independent Rademacher variables uniformly distributed over $\{\pm 1\}$ and $\boldsymbol{v} = (v_{i,j})_{i,j}$ is a sequence of random variables distributed according to $\mathcal{U}$. Note that $v_{i,j}$s themselves do not need to be independent.*

*Proof.* Fix a sample $S = (x_1, \ldots, x_m)$. Then, we can rewrite the left-hand side of (11) as follows:

$$
\frac{1}{m} \mathop{\mathbb{E}}_{\boldsymbol{\sigma}} \left[ \sup_{\mathbf{h} \in \mathcal{H}} \sum_{i=1}^m \sigma_i \Psi_i(\mathbf{h}(x_i)) \right] = \frac{1}{m} \mathop{\mathbb{E}}_{\sigma_1, \ldots, \sigma_{m-1}} \left[ \mathop{\mathbb{E}}_{\sigma_m} \left[ \sup_{\mathbf{h} \in \mathcal{H}} U_{m-1}(\mathbf{h}) + \sigma_m \Psi_m(\mathbf{h}(x_m)) \right] \right],
$$

where $U_{m-1}(\mathbf{h}) = \sum_{i=1}^{m-1} \sigma_i \Psi_i(\mathbf{h}(x_i))$. Assume that the suprema can be attained and let $\mathbf{h}_1, \mathbf{h}_2 \in \mathcal{H}$ be the hypotheses satisfying

$$
U_{m-1}(\mathbf{h}_1) + \Psi_m(\mathbf{h}_1(x_m)) = \sup_{\mathbf{h} \in \mathcal{H}} U_{m-1}(\mathbf{h}) + \Psi_m(\mathbf{h}(x_m))
$$

$$
U_{m-1}(\mathbf{h}_2) - \Psi_m(\mathbf{h}_2(x_m)) = \sup_{\mathbf{h} \in \mathcal{H}} U_{m-1}(\mathbf{h}) - \Psi_m(\mathbf{h}(x_m)).
$$

When the suprema are not reached, a similar argument to what follows can be given by considering instead hypotheses that are $\epsilon$-close to the suprema for any $\epsilon > 0$. By definition of expectation, since $\sigma_m$ is uniformly distributed over $\{\pm 1\}$, we can write

$$
\mathop{\mathbb{E}}_{\sigma_m} \left[ \sup_{\mathbf{h} \in \mathcal{H}} U_{m-1}(\mathbf{h}) + \sigma_m \Psi_m(\mathbf{h}_1(x_m)) \right]
$$
$$
= \frac{1}{2} \sup_{\mathbf{h} \in \mathcal{H}} U_{m-1}(\mathbf{h}) + \Psi_m(\mathbf{h}_1(x_m)) + \frac{1}{2} \sup_{\mathbf{h} \in \mathcal{H}} U_{m-1}(\mathbf{h}) - \Psi_m(\mathbf{h}(x_m))
$$
$$
= \frac{1}{2}[U_{m-1}(\mathbf{h}_1) + \Psi_m(\mathbf{h}_1(x_m))] + \frac{1}{2}[U_{m-1}(\mathbf{h}_2) - \Psi_m(\mathbf{h}_2(x_m))].
$$

Next, using the $\mu_m$-Lipschitzness of $\Psi_m$ and the Khintchine-Kahane inequality, we can write

$$
\mathop{\mathbb{E}}_{\sigma_m} \left[ \sup_{\mathbf{h} \in \mathcal{H}} U_{m-1}(\mathbf{h}) + \sigma_m (\Psi_m \circ h)(x_m) \right]
$$
$$
\leq \frac{1}{2}[U_{m-1}(\mathbf{h}_1) + U_{m-1}(\mathbf{h}_2) + \mu_m \|\mathbf{h}_1(x_m) - \mathbf{h}_2(x_m)\|_{\infty,2}]
$$
$$
\leq \frac{1}{2}\left[ U_{m-1}(\mathbf{h}_1) + U_{m-1}(\mathbf{h}_2) + \mu_m\sqrt{2} \mathop{\mathbb{E}}_{\epsilon_{m1},\ldots,\epsilon_{mc}} \left[ \Big| \sum_{j=1}^{c} \epsilon_{mj}\|h_{1,j}(x_m,\cdot) - h_{2,j}(x_m,\cdot)\|_{\infty} \Big| \right] \right].
$$

Define the random variables $\upsilon_{mj} = \upsilon_{mj}(\boldsymbol{\sigma}) = \mathrm{argmax}_{k \in [d]} |h_{1,j}(x_m,k) - h_{2,j}(x_m,k)|$.

Now, let $\boldsymbol{\epsilon}_m$ denote $(\epsilon_{m1},\ldots,\epsilon_{mc})$ and let $s(\boldsymbol{\epsilon}_m) \in \{\pm 1\}$ denote the sign of $\sum_{j=1}^{c} \epsilon_{mj}\|h_{1,j}(x_m,\cdot) - h_{2,j}(x_m,\cdot)\|_{\infty}$. Then, the following holds:

$$
\mathop{\mathbb{E}}_{\sigma_m} \left[ \sup_{\mathbf{h} \in \mathcal{H}} U_{m-1}(\mathbf{h}) + \sigma_m (\Psi_m \circ h)(x_m) \right]
$$
$$
\leq \frac{1}{2} \mathop{\mathbb{E}}_{\boldsymbol{\epsilon}_m} \left[ U_{m-1}(\mathbf{h}_1) + U_{m-1}(\mathbf{h}_2) + \mu_m\sqrt{2} \Big| \sum_{j=1}^{c} \epsilon_{mj}\|h_{1,j}(x_m,\cdot) - h_{2,j}(x_m,\cdot)\|_{\infty} \Big| \right]
$$
$$
\leq \frac{1}{2} \mathop{\mathbb{E}}_{\boldsymbol{\epsilon}_m} \Big[ U_{m-1}(\mathbf{h}_1) + U_{m-1}(\mathbf{h}_2)
$$
$$
\qquad + \mu_m\sqrt{2}\, s(\boldsymbol{\epsilon}_m) \sum_{j=1}^{c} \epsilon_{mj}|h_{1,j}(x_m,\upsilon_{mj}) - h_{2,j}(x_m,\upsilon_{mj})| \Big]
$$
$$
= \frac{1}{2} \mathop{\mathbb{E}}_{\boldsymbol{\epsilon}_m} \Big[ U_{m-1}(\mathbf{h}_1) + U_{m-1}(\mathbf{h}_2)
$$
$$
\qquad + \mu_m\sqrt{2}\, s(\boldsymbol{\epsilon}_m) \sum_{j=1}^{c} \epsilon_{mj}(h_{1,j}(x_m,\upsilon_{mj}) - h_{2,j}(x_m,\upsilon_{mj})) \Big]
$$
$$
= \frac{1}{2} \mathop{\mathbb{E}}_{\boldsymbol{\epsilon}_m} \Big[ U_{m-1}(\mathbf{h}_1) + \mu_m\sqrt{2}\, s(\boldsymbol{\epsilon}_m) \sum_{j=1}^{c} \epsilon_{mj}h_{1,j}(x_m,\upsilon_{mj})
$$
$$
\qquad + U_{m-1}(\mathbf{h}_2) - \mu_m\sqrt{2}\, s(\boldsymbol{\epsilon}_m) \sum_{j=1}^{c} \epsilon_{mj}h_{2,j}(x_m,\upsilon_{mj}) \Big].
$$

After taking expectation over $\boldsymbol{v}$, the rest of the proof proceeds the same way as the argument in Lemma 5:

$$\frac{1}{2} \mathop{\mathbb{E}}_{\boldsymbol{v}\sim\mathcal{U},\boldsymbol{\epsilon}_m} \left[ U_{m-1}(\mathbf{h}_1) + \mu_m\sqrt{2}\, s(\boldsymbol{\epsilon}_m) \sum_{j=1}^{c} \epsilon_{mj} h_{1,j}(x_m, v_{mj}) \right.$$

$$\left. + U_{m-1}(\mathbf{h}_2) - \mu_m\sqrt{2}\, s(\boldsymbol{\epsilon}_m) \sum_{j=1}^{c} \epsilon_{mj} h_{2,j}(x_m, v_{mj}) \right]$$

$$\leq \frac{1}{2} \mathop{\mathbb{E}}_{\boldsymbol{v}\sim\mathcal{U},\boldsymbol{\epsilon}_m} \left[ \sup_{\mathbf{h}\in\mathcal{H}} \left( U_{m-1}(\mathbf{h}) + \mu_m\sqrt{2}\, s(\boldsymbol{\epsilon}_m) \sum_{j=1}^{c} \epsilon_{mj} h_j(x_m, v_{mj}) \right) \right.$$

$$\left. + \sup_{\mathbf{h}\in\mathcal{H}} \left( U_{m-1}(\mathbf{h}) - \mu_m\sqrt{2}\, s(\boldsymbol{\epsilon}_m) \sum_{j=1}^{c} \epsilon_{mj} h_j(x_m, v_{mj}) \right) \right]$$

$$= \mathop{\mathbb{E}}_{\boldsymbol{v}\sim\mathcal{U},\boldsymbol{\epsilon}_m} \left[ \mathop{\mathbb{E}}_{\sigma_m} \left[ \sup_{\mathbf{h}\in\mathcal{H}} U_{m-1}(\mathbf{h}) + \mu_m\sqrt{2}\, \sigma_m \sum_{j=1}^{c} \epsilon_{mj} h_j(x_m, v_{mj}) \right] \right]$$

$$= \mathop{\mathbb{E}}_{\boldsymbol{v}\sim\mathcal{U},\boldsymbol{\epsilon}_m} \left[ \sup_{\mathbf{h}\in\mathcal{H}} U_{m-1}(\mathbf{h}) + \mu_m\sqrt{2} \sum_{j=1}^{c} \epsilon_{mj} h_j(x_m, v_{mj}) \right],$$

Proceeding in the same way for all other $\sigma_i$s ($i < m$) completes the proof. $\qquad\square$

### A.3 General structured prediction learning bounds

In this section, we give the proof of several general structured prediction bounds in terms of the notion of factor graph Rademacher complexity. We will use the additive and multiplicative margin losses of a hypothesis $h$, which are the population versions of the empirical margin losses we introduced in (5) and (6) and are defined as follows:

$$R_\rho^{\mathrm{add}}(h) = \mathop{\mathbb{E}}_{(x,y)\sim\mathcal{D}} \left[ \Phi^* \left( \max_{y'\neq y} \mathsf{L}(y',y) - \tfrac{1}{\rho}\big[h(x,y) - h(x,y')\big] \right) \right]$$

$$R_\rho^{\mathrm{mult}}(h) = \mathop{\mathbb{E}}_{(x,y)\sim\mathcal{D}} \left[ \Phi^* \left( \max_{y'\neq y} \mathsf{L}(y',y)\left(1 - \tfrac{1}{\rho}[h(x,y) - h(x,y')]\right) \right) \right].$$

The following is our general margin bound for structured prediction.

**Theorem 1.** *Fix $\rho > 0$. For any $\delta > 0$, with probability at least $1 - \delta$ over the draw of a sample $S$ of size $m$, the following holds for all $h \in \mathcal{H}$,*

$$R(h) \leq R_\rho^{add}(h) \leq \widehat{R}_{S,\rho}^{add}(h) + \frac{4\sqrt{2}}{\rho}\mathfrak{R}_m^G(\mathcal{H}) + M\sqrt{\frac{\log\frac{1}{\delta}}{2m}},$$

$$R(h) \leq R_\rho^{mult}(h) \leq \widehat{R}_{S,\rho}^{mult}(h) + \frac{4\sqrt{2}M}{\rho}\mathfrak{R}_m^G(\mathcal{H}) + M\sqrt{\frac{\log\frac{1}{\delta}}{2m}}.$$

*Proof.* Let $\Phi_u(v) = \Phi^*(u - \frac{v}{\rho})$, where $\Phi^*(r) = \min(M, \max(0, r))$. Observe that for any $u \in [0, M]$, $u\mathbf{1}_{v\leq 0} \leq \Phi_u(v)$ for all $v$. Therefore, by Lemma 4 and monotonicity of $\Phi^*$,

$$R(h) \leq \mathop{\mathbb{E}}_{(x,y)\sim\mathcal{D}} [\max_{y'\neq y} \Phi_{\mathsf{L}(y',y)}(h(x,y) - h(x,y'))]$$

$$= \mathop{\mathbb{E}}_{(x,y)\sim\mathcal{D}} \left[ \Phi^* \left( \max_{y'\neq y} \left( \mathsf{L}(y',y) - \frac{h(x,y) - h(x,y')}{\rho} \right) \right) \right]$$

$$= R_\rho^{\mathrm{add}}(h).$$

Define

$$\mathcal{H}_0 = \left\{ (x,y) \mapsto \Phi^* \Big( \max_{y' \neq y} \Big( \mathsf{L}(y',y) - \frac{h(x,y) - h(x,y')}{\rho} \Big) \Big) : h \in \mathcal{H} \right\},$$

$$\mathcal{H}_1 = \left\{ (x,y) \mapsto \max_{y' \neq y} \Big( \mathsf{L}(y',y) - \frac{h(x,y) - h(x,y')}{\rho} \Big) : h \in \mathcal{H} \right\}.$$

By standard Rademacher complexity bounds (Koltchinskii and Panchenko [2002]), for any $\delta > 0$, with probability at least $1 - \delta$, the following inequality holds for all $h \in \mathcal{H}$:

$$R_\rho^{\mathrm{add}}(h) \leq \widehat{R}_{S,\rho}^{\mathrm{add}}(h) + 2\mathfrak{R}_m(\mathcal{H}_0) + M\sqrt{\frac{\log \frac{1}{\delta}}{2m}},$$

where $\mathfrak{R}_m(\mathcal{H}_0)$ is the Rademacher complexity of the family $\mathcal{H}_0$:

$$\mathfrak{R}_m(\mathcal{H}_0) = \frac{1}{m} \mathop{\mathbb{E}}_{S \sim \mathcal{D}^m} \mathop{\mathbb{E}}_{\boldsymbol{\sigma}} \left[ \sup_{h \in \mathcal{H}} \sum_{i=1}^m \sigma_i \Phi^* \Big( \max_{y' \neq y_i} \Big( \mathsf{L}(y', y_i) - \frac{h(x_i, y_i) - h(x_i, y')}{\rho} \Big) \Big) \right]$$

and where $\boldsymbol{\sigma} = (\sigma_1, \ldots, \sigma_m)$ with $\sigma_i$s independent Rademacher random variables uniformly distributed over $\{\pm 1\}$. Since $\Phi^*$ is 1-Lipschitz, by Talagrand's contraction lemma (Ledoux and Talagrand [1991], Mohri et al. [2012]), we have $\widehat{\mathfrak{R}}_S(\mathcal{H}_0) \leq \widehat{\mathfrak{R}}_S(\mathcal{H}_1)$. By taking an expectation over $S$, this inequality carries over to the true Rademacher complexities as well. Now, observe that by the sub-additivity of the supremum, the following holds:

$$\widehat{\mathfrak{R}}_S(\mathcal{H}_1) \leq \frac{1}{m} \mathop{\mathbb{E}}_{\boldsymbol{\sigma}} \left[ \sup_{h \in \mathcal{H}} \sum_{i=1}^m \sigma_i \max_{y' \neq y_i} \Big( \mathsf{L}(y', y_i) + \frac{h(x_i, y')}{\rho} \Big) \right]$$
$$+ \frac{1}{m} \mathop{\mathbb{E}}_{\boldsymbol{\sigma}} \left[ \sup_{h \in \mathcal{H}} \sum_{i=1}^m \sigma_i \frac{h(x_i, y_i)}{\rho} \right],$$

where we also used for the last term the fact that $-\sigma_i$ and $\sigma_i$ admit the same distribution. We use Lemma 5 to bound each of the two terms appearing on the right-hand side separately. To do so, we we first show the Lipschitzness of $h \mapsto \max_{y' \neq y_i} \Big( \mathsf{L}(y', y_i) + \frac{h(x_i, y')}{\rho} \Big)$. Observe that the following chain of inequalities holds for any $h, \widetilde{h} \in \mathcal{H}$:

$$\left| \max_{y \neq y_i} \Big( \mathsf{L}(y, y_i) + \frac{h(x_i, y)}{\rho} \Big) - \max_{y \neq y_i} \Big( \mathsf{L}(y, y_i) + \frac{\widetilde{h}(x_i, y)}{\rho} \Big) \right|$$

$$\leq \frac{1}{\rho} \max_{y \neq y_i} \left| h(x_i, y) - \widetilde{h}(x_i, y) \right|$$

$$\leq \frac{1}{\rho} \max_{y \in \mathcal{Y}} \left| h(x_i, y) - \widetilde{h}(x_i, y) \right|$$

$$= \frac{1}{\rho} \max_{y \in \mathcal{Y}} \left| \sum_{f \in F_i} (h_f(x_i, y_f) - \widetilde{h}_f(x_i, y_f)) \right|$$

$$\leq \frac{1}{\rho} \sum_{f \in F_i} \max_{y \in \mathcal{Y}} \left| (h_f(x_i, y_f) - \widetilde{h}_f(x_i, y_f)) \right|$$

$$= \frac{1}{\rho} \sum_{f \in F_i} \max_{y \in \mathcal{Y}_f} \left| (h_f(x_i, y) - \widetilde{h}_f(x_i, y)) \right|$$

$$\leq \frac{\sqrt{|F_i|}}{\rho} \sqrt{\sum_{f \in F_i} \Big[ \max_{y \in \mathcal{Y}_f} |(h_f(x_i, y) - \widetilde{h}_f(x_i, y))| \Big]^2}$$

$$= \frac{\sqrt{|F_i|}}{\rho} \sqrt{\sum_{f \in F_i} \max_{y \in \mathcal{Y}_f} |(h_f(x_i, y) - \widetilde{h}_f(x_i, y))|^2}$$

$$\leq \frac{\sqrt{|F_i|}}{\rho} \sqrt{\sum_{f \in F_i} \sum_{y \in \mathcal{Y}_f} |(h_f(x_i, y) - \widetilde{h}_f(x_i, y))|^2}.$$

We can therefore apply Lemma 5, which yields

$$\frac{1}{m}\mathop{\mathbb{E}}_{\boldsymbol{\sigma}}\left[\sup_{h\in\mathcal{H}}\sum_{i=1}^{m}\sigma_i\max_{y'\neq y_i}\left(\mathsf{L}(y',y_i)+\frac{h(x_i,y')}{\rho}\right)\right]$$

$$\leq\frac{\sqrt{2}}{m}\mathop{\mathbb{E}}_{\boldsymbol{\epsilon}}\left[\sup_{h\in\mathcal{H}}\sum_{i=1}^{m}\sum_{f\in F_i}\sum_{y\in\mathcal{Y}_f}\epsilon_{i,f,y}\frac{\sqrt{|F_i|}}{\rho}h_f(x_i,y)\right]=\frac{\sqrt{2}}{\rho}\widehat{\mathfrak{R}}_S^G(\mathcal{H}).$$

Similarly, for the second term, observe that the following Lipschitz property holds:

$$\left|\frac{h(x_i,y_i)}{\rho}-\frac{\widetilde{h}(x_i,y_i)}{\rho}\right|\leq\frac{1}{\rho}\max_{y\in\mathcal{Y}}\left|h(x_i,y)-\widetilde{h}(x_i,y)\right|$$

$$\leq\frac{\sqrt{|F_i|}}{\rho}\sqrt{\sum_{f\in F_i}\sum_{y\in\mathcal{Y}}|(h_f(x_i,y)-\widetilde{h}_f(x_i,y))|^2}.$$

We can therefore apply Lemma 5 and obtain the following:

$$\frac{1}{m}\mathop{\mathbb{E}}_{\boldsymbol{\sigma}}\left[\sup_{h\in\mathcal{H}}\sum_{i=1}^{m}\sigma_i\frac{h(x_i,y_i)}{\rho}\right]\leq\frac{\sqrt{2}}{m}\mathop{\mathbb{E}}_{\boldsymbol{\epsilon}}\left[\sup_{h\in\mathcal{H}}\sum_{i=1}^{m}\sum_{f\in F_i}\sum_{y\in\mathcal{Y}_f}\epsilon_{i,f,y}\frac{\sqrt{|F_i|}}{\rho}h_f(x_i,y)\right]=\frac{\sqrt{2}}{\rho}\widehat{\mathfrak{R}}_S^G(\mathcal{H}).$$

Taking the expectation over $S$ of the two inequalities shows that $\mathfrak{R}_m(\mathcal{H}_1)\leq\frac{2\sqrt{2}}{\rho}\mathfrak{R}_m^G(\mathcal{H})$, which completes the proof of the first statement.

The second statement can be proven in a similar way with $\Phi_u(v)=\Phi^*(u(1-\frac{v}{\rho}))$. In particular, by standard Rademacher complexity bounds, McDiarmid's inequality, and Talagrand's contraction lemma, we can write

$$R_\rho^{\mathrm{mult}}(h)\leq\widehat{R}_{S,\rho}^{\mathrm{mult}}(h)+2\mathfrak{R}_m(\widetilde{\mathcal{H}}_1)+M\sqrt{\frac{\log\frac{1}{\delta}}{2m}},$$

where

$$\widetilde{\mathcal{H}}_1=\left\{(x,y)\mapsto\max_{y'\neq y}\mathsf{L}(y',y)\left(1-\frac{h(x,y)-h(x,y')}{\rho}\right):h\in\mathcal{H}\right\}.$$

We observe that the following inequality holds:

$$\left|\max_{y\neq y_i}\mathsf{L}(y,y_i)\left(1-\frac{h(x_i,y_i)-h(x_i,y)}{\rho}\right)-\max_{y\neq y_i}\mathsf{L}(y,y_i)\left(1-\frac{\widetilde{h}(x_i,y_i)-\widetilde{h}(x_i,y)}{\rho}\right)\right|$$

$$\leq\frac{2M}{\rho}\max_{y\in\mathcal{Y}}\left|h(x_i,y)-\widetilde{h}(x_i,y)\right|.$$

Then, the rest of the proof follows from Lemma 5 as in the previous argument. $\qquad\square$

In the proof above, we could have applied McDiarmid's inequality to bound the Rademacher complexity of $\mathcal{H}_0$ by its empirical counterpart at the cost of slightly increasing the exponential concentration term:

$$R_\rho^{\mathrm{add}}(h)\leq\widehat{R}_{S,\rho}^{\mathrm{add}}(h)+2\widehat{\mathfrak{R}}_S(\mathcal{H}_0)+3M\sqrt{\frac{\log\frac{1}{\delta}}{2m}}.$$

Since Talagrand's contraction lemma holds for empirical Rademacher complexities and the remainder of the proof involves bounding the empirical Rademacher complexity of $\mathcal{H}_1$ before taking an expectation over the sample at the end, we can apply the same arguments without the final expectation to arrive at the following analogue of Theorem 1 in terms of empirical complexities:

**Theorem 7.** *Fix $\rho > 0$. For any $\delta > 0$, with probability at least $1 - \delta$ over the draw of a sample $S$ of size $m$, the following holds for all $h \in \mathcal{H}$,*

$$R(h) \leq R_\rho^{add}(h) \leq \widehat{R}_{S,\rho}^{add}(h) + \frac{4\sqrt{2}}{\rho}\widehat{\mathfrak{R}}_S^G(\mathcal{H}) + 3M\sqrt{\frac{\log\frac{1}{\delta}}{2m}},$$

$$R(h) \leq R_\rho^{mult}(h) \leq \widehat{R}_{S,\rho}^{mult}(h) + \frac{4\sqrt{2}M}{\rho}\widehat{\mathfrak{R}}_S^G(\mathcal{H}) + 3M\sqrt{\frac{\log\frac{1}{\delta}}{2m}}.$$

This theorem will be useful for many of our applications, which are based on bounding the empirical factor graph Rademacher complexity for different hypothesis classes.

## A.4 Concentration of the empirical factor graph Rademacher complexity

In this section, we show that, as with the standard notion of Rademacher complexity, the empirical factor graph Rademacher complexity also concentrates around its mean.

**Lemma 8.** *Let $\mathcal{H}$ be a family of scoring functions mapping $\mathcal{X} \times \mathcal{Y} \to \mathbb{R}$ bounded by a constant $C$. Let $S$ be a training sample of size $m$ drawn i.i.d. according to some distribution $\mathcal{D}$ on $\mathcal{X} \times \mathcal{Y}$, and let $\mathcal{D}_\mathcal{X}$ be the marginal distribution on $\mathcal{X}$. For any point $x \in \mathcal{X}$, let $F_x$ denote its associated set of factor nodes. Then, with probability at least $1 - \delta$ over the draw of sample $S \sim \mathcal{D}^m$,*

$$\left|\widehat{\mathfrak{R}}_S^G(\mathcal{H}) - \mathfrak{R}_m^G(\mathcal{H})\right| \leq 2C \sup_{x \in supp(\mathcal{D}_\mathcal{X})} \sum_{f \in F_x} |\mathcal{Y}_f|\sqrt{|F_x|}\sqrt{\frac{\log\frac{2}{\delta}}{2m}}.$$

*Proof.* Let $S = (x_1, x_2, \ldots, x_m)$ and $S' = (x'_1, x'_2, \ldots, x'_m)$ be two samples differing by one point $x_j$ and $x'_j$ (i.e. $x_i = x'_i$ for $i \neq j$). Then

$$
\begin{aligned}
\widehat{\mathfrak{R}}_S^G(\mathcal{H}) - \widehat{\mathfrak{R}}_{S'}^G(\mathcal{H}) &\leq \frac{1}{m}\mathbb{E}_\epsilon\left[\sup_{h \in \mathcal{H}} \sum_{i=1}^m \sum_{f \in F_i} \sum_{y \in \mathcal{Y}_f} \sqrt{|F_i|}\,\epsilon_{i,f,y}\, h_f(x_i, y)\right] \\
&\quad - \frac{1}{m}\mathbb{E}_\epsilon\left[\sup_{h \in \mathcal{H}} \sum_{i=1}^m \sum_{f \in F_i} \sum_{y \in \mathcal{Y}_f} \sqrt{|F_i|}\,\epsilon_{i,f,y}\, h_f(x'_i, y)\right] \\
&= \frac{1}{m}\mathbb{E}_\epsilon\left[\sup_{h \in \mathcal{H}} \sum_{f \in F_{x_j}} \sum_{y \in \mathcal{Y}_f} \sqrt{|F_j|}\,\epsilon_{j,f,y}\, h_f(x_j, y)\right. \\
&\quad \left. - \sum_{f' \in F_{x'_j}} \sum_{y \in \mathcal{Y}'_f} \sqrt{|F_{x'_j}|}\,\epsilon_{j,f',y}\, h_f(x'_j, y)\right] \\
&\leq \frac{2}{m}\sup_{x \in supp(\mathcal{D}_\mathcal{X})} \sup_{h \in \mathcal{H}} \sum_{f \in F_x} \sum_{y \in \mathcal{Y}_f} \sqrt{|F_x|}|h_f(x, y)|.
\end{aligned}
$$

The same upper bound also holds for $\widehat{\mathfrak{R}}_{S'}^G(\mathcal{H}) - \widehat{\mathfrak{R}}_S^G(\mathcal{H})$. The result now follows from McDiarmid's inequality. $\square$

## A.5 Bounds on the factor graph Rademacher complexity

The following lemma is a standard bound on the expectation of the maximum of $n$ zero-mean bounded random variables, which will be used in the proof of our bounds on factor graph Rademacher complexity.

**Lemma 9.** *Let $X_1 \ldots X_n$ be $n \geq 1$ real-valued random variables such that for all $j \in [1, n]$, $X_j = \sum_{i=1}^{m_j} Y_{ij}$ where, for each fixed $j \in [1, n]$, $Y_{ij}$ are independent zero mean random variables with $|Y_{ij}| \leq t_{ij}$. Then, the following inequality holds:*

$$\mathbb{E}\left[\max_{j \in [1,n]} X_j\right] \leq t\sqrt{2\log n},$$

*with* $t = \sqrt{\max_{j \in [1,n]} \sum_{i=1}^{m_j} t_{ij}^2}$.

The following are upper bounds on the factor graph Rademacher complexity for $\mathcal{H}_1$ and $\mathcal{H}_2$, as defined in Section 3. Similar guarantees can be given for other hypothesis sets $\mathcal{H}_p$ with $p > 1$.

**Theorem 2.** *For any sample* $S = (x_1, \ldots, x_m)$, *the following upper bounds hold for the empirical factor graph complexity of* $\mathcal{H}_1$ *and* $\mathcal{H}_2$:

$$\widehat{\mathfrak{R}}_S^G(\mathcal{H}_1) \leq \frac{\Lambda_1 r_\infty}{m} \sqrt{s \log(2N)}, \qquad \widehat{\mathfrak{R}}_S^G(\mathcal{H}_2) \leq \frac{\Lambda_2 r_2}{m} \sqrt{\sum_{i=1}^m \sum_{f \in F_i} \sum_{y \in \mathcal{Y}_f} |F_i|},$$

*where* $r_\infty = \max_{i,f,y} \|\Psi_f(x_i, y)\|_\infty$, $r_2 = \max_{i,f,y} \|\Psi_f(x_i, y)\|_2$ *and where* $s$ *is a sparsity factor defined by* $s = \max_{j \in [1,N]} \sum_{i=1}^m \sum_{f \in F_i} \sum_{y \in \mathcal{Y}_f} |F_i| 1_{\Psi_j(x_i, y) \neq 0}$.

*Proof.* By definition of the dual norm and Lemma 9 (or Massart's lemma), the following holds:

$$m\widehat{\mathfrak{R}}_S^G(\mathcal{H}_1) = \mathbb{E}_{\boldsymbol{\epsilon}} \left[ \sup_{\|\mathbf{w}\|_1 \leq \Lambda_1} \mathbf{w} \cdot \sum_{i=1}^m \sum_{f \in F_i} \sum_{y \in \mathcal{Y}_f} \sqrt{|F_i|} \epsilon_{i,f,y} \boldsymbol{\Psi}_f(x_i, y) \right]$$

$$= \Lambda_1 \mathbb{E}_{\boldsymbol{\epsilon}} \left[ \left\| \sum_{i=1}^m \sum_{f \in F_i} \sum_{y \in \mathcal{Y}_f} \sqrt{|F_i|} \epsilon_{i,f,y} \boldsymbol{\Psi}_f(x_i, y) \right\|_\infty \right]$$

$$= \Lambda_1 \mathbb{E}_{\boldsymbol{\epsilon}} \left[ \max_{j \in [1,N], \sigma \in \{-1,+1\}} \sigma \sum_{i=1}^m \sum_{f \in F_i} \sum_{y \in \mathcal{Y}_f} \sqrt{|F_i|} \epsilon_{i,f,y} \Psi_{f,j}(x_i, y) \right]$$

$$= \Lambda_1 \mathbb{E}_{\boldsymbol{\epsilon}} \left[ \max_{j \in [1,N], \sigma \in \{-1,+1\}} \sigma \sum_{i=1}^m \sum_{f \in F_i} \sum_{y \in \mathcal{Y}_f} \sqrt{|F_i|} \epsilon_{i,f,y} \Psi_{f,j}(x_i, y) 1_{\Psi_{f,j}(x_i,y) \neq 0} \right]$$

$$\leq \Lambda_1 \sqrt{2 \left( \max_{j \in [1,N]} \sum_{i=1}^m \sum_{f \in F_i} \sum_{y \in \mathcal{Y}_f} |F_i| 1_{\Psi_j(x_i,y) \neq 0} \right) r_\infty^2 \log(2N)}$$

$$= \Lambda_1 r_\infty \sqrt{2s \log(2N)},$$

which completes the proof of the first statement. The second statement can be proven in a similar way using the the definition of the dual norm and Jensen's inequality:

$$m\widehat{\mathfrak{R}}_S^G(\mathcal{H}_2) = \mathbb{E}_{\boldsymbol{\epsilon}} \left[ \sup_{\|\mathbf{w}\|_2 \leq \Lambda_2} \mathbf{w} \cdot \sum_{i=1}^m \sum_{f \in F_i} \sum_{y \in \mathcal{Y}_f} \sqrt{|F_i|} \epsilon_{i,f,y} \boldsymbol{\Psi}_f(x_i, y) \right]$$

$$= \Lambda_2 \mathbb{E}_{\boldsymbol{\epsilon}} \left[ \left\| \sum_{i=1}^m \sum_{f \in F_i} \sum_{y \in \mathcal{Y}_f} \sqrt{|F_i|} \epsilon_{i,f,y} \boldsymbol{\Psi}_f(x_i, y) \right\|_2 \right]$$

$$= \Lambda_2 \left( \mathbb{E}_{\boldsymbol{\epsilon}} \left[ \left\| \sum_{i=1}^m \sum_{f \in F_i} \sum_{y \in \mathcal{Y}_f} \sqrt{|F_i|} \epsilon_{i,f,y} \boldsymbol{\Psi}_f(x_i, y) \right\|_2^2 \right] \right)^{\frac{1}{2}}$$

$$= \Lambda_2 \left( \sum_{i=1}^m \sum_{f \in F_i} \sum_{y \in \mathcal{Y}_f} |F_i| \|\boldsymbol{\Psi}_f(x_i, y)\|_2^2 \right)^{\frac{1}{2}}$$

$$\leq \Lambda_2 r_2 \sqrt{\sum_{i=1}^m \sum_{f \in F_i} \sum_{y \in \mathcal{Y}_f} |F_i|},$$

which concludes the proof. $\square$

### A.6 Learning guarantees for structured prediction with linear hypotheses

The following result is a direct consequence of Theorem 7 and Theorem 2.

**Corollary 10.** *Fix $\rho > 0$. For any $\delta > 0$, with probability at least $1 - \delta$ over the draw of a sample $S$ of size $m$, the following holds for all $h \in \mathcal{H}_1$,*

$$R(h) \leq \widehat{R}_{S,\rho}^{add}(h) + \frac{4\sqrt{2}}{\rho m}\Lambda_1 r_\infty \sqrt{s\log(2N)} + 3M\sqrt{\frac{\log\frac{2}{\delta}}{2m}},$$

$$R(h) \leq \widehat{R}_{S,\rho}^{mult}(h) + \frac{4\sqrt{2}M}{\rho m}\Lambda_1 r_\infty \sqrt{s\log(2N)} + 3M\sqrt{\frac{\log\frac{2}{\delta}}{2m}}.$$

*Similarly, for any $\delta > 0$, with probability at least $1 - \delta$ over the draw of a sample $S$ of size $m$, the following holds for all $h \in \mathcal{H}_2$,*

$$R(h) \leq \widehat{R}_{S,\rho}^{add}(h) + \frac{4\sqrt{2}}{\rho m}\Lambda_2 r_2 \sqrt{\sum_{i=1}^{m}\sum_{f\in F_i}\sum_{y\in\mathcal{Y}_f}|F_i|} + 3M\sqrt{\frac{\log\frac{2}{\delta}}{2m}},$$

$$R(h) \leq \widehat{R}_{S,\rho}^{mult}(h) + \frac{4\sqrt{2}M}{\rho m}\Lambda_2 r_2 \sqrt{\sum_{i=1}^{m}\sum_{f\in F_i}\sum_{y\in\mathcal{Y}_f}|F_i|} + 3M\sqrt{\frac{\log\frac{2}{\delta}}{2m}}.$$

### A.7 Learning guarantees for multi-class classification with linear hypotheses

The following result is a direct consequence of Corollary 10 and the observation that for multi-class classification $|F_i| = 1$ and $d_i = \max_{f\in F_i}|\mathcal{Y}_f| = c$. Note that our multi-class learning guarantees hold for arbitrary bounded losses. To the best of our knowledge this is a novel result in this setting. In particular, these guarantees apply to the special case of the standard multi-class zero-one loss $L(y, y') = 1_{\{y\neq y'\}}$ which is bounded by $M = 1$.

**Corollary 11.** *Fix $\rho > 0$. For any $\delta > 0$, with probability at least $1 - \delta$ over the draw of a sample $S$ of size $m$, the following holds for all $h \in \mathcal{H}_1$,*

$$R(h) \leq \widehat{R}_{S,\rho}^{add}(h) + \frac{4\sqrt{2}\Lambda_1 r_\infty}{\rho}\sqrt{\frac{c\log(2N)}{m}} + 3M\sqrt{\frac{\log\frac{2}{\delta}}{2m}},$$

$$R(h) \leq \widehat{R}_{S,\rho}^{mult}(h) + \frac{4\sqrt{2}\Lambda_1 r_\infty}{\rho}\sqrt{\frac{c\log(2N)}{m}} + 3M\sqrt{\frac{\log\frac{2}{\delta}}{2m}}.$$

*Similarly, for any $\delta > 0$, with probability at least $1 - \delta$ over the draw of a sample $S$ of size $m$, the following holds for all $h \in \mathcal{H}_2$,*

$$R(h) \leq \widehat{R}_{S,\rho}^{add}(h) + \frac{4\sqrt{2}\Lambda_2 r_2}{\rho}\sqrt{\frac{c}{m}} + 3M\sqrt{\frac{\log\frac{2}{\delta}}{2m}},$$

$$R(h) \leq \widehat{R}_{S,\rho}^{mult}(h) + \frac{4\sqrt{2}\Lambda_2 r_2}{\rho}\sqrt{\frac{c}{m}} + 3M\sqrt{\frac{\log\frac{2}{\delta}}{2m}}.$$

Consider the following set of linear hypothesis:
$$\mathcal{H}_{2,1} = \{x \mapsto \mathbf{w}\cdot\mathbf{\Psi}(x, y)\colon \|\mathbf{w}\|_{2,1} \leq \Lambda_{2,1}, y \in [c]\},$$
where $\mathbf{\Psi}(x, y) = (0, \ldots 0, \mathbf{\Psi}_y(x), 0, \ldots, 0)^T \in \mathbb{R}^{N_1 \times \ldots, N_c}$ and $\mathbf{w} = (\mathbf{w}_1, \ldots, \mathbf{w}_c)$ with $\|\mathbf{w}\|_{2,1} = \sum_{y=1}^{c}\|\mathbf{w}_y\|_2$. In this case, $\mathbf{w}\cdot\mathbf{\Psi}(x, y) = \mathbf{w}_y\cdot\mathbf{\Psi}_y(x)$. The standard scenario in multi-class classification is when $\mathbf{\Psi}_y(x) = \mathbf{\Psi}(x)$ is the same for all $y$.

**Corollary 12.** *Fix $\rho > 0$. For any $\delta > 0$, with probability at least $1 - \delta$ over the draw of a sample $S$ of size $m$, the following holds for all $h \in \mathcal{H}_{2,1}$,*

$$R(h) \leq \widehat{R}_{S,\rho}^{add}(h) + \frac{16\Lambda_{2,1} r_{2,\infty}(\log(c))^{1/4}}{\rho\sqrt{m}} + 3M\sqrt{\frac{\log\frac{2}{\delta}}{2m}},$$

$$R(h) \leq \widehat{R}_{S,\rho}^{mult}(h) + \frac{16\Lambda_{2,1} r_{2,\infty}(\log(c))^{1/4}}{\rho\sqrt{m}} + 3M\sqrt{\frac{\log\frac{2}{\delta}}{2m}},$$

*where $r_{2,\infty} = \max_{i,y}\|\mathbf{\Psi}_y(x_i)\|_2$.*

*Proof.* By definition of the dual norm and $\mathcal{H}_{2,1}$, the following holds:

$$m\widehat{\mathfrak{R}}_S^G(\mathcal{H}_{2,1}) = \mathbb{E}_{\boldsymbol{\epsilon}}\left[\sup_{\|\mathbf{w}\|_{2,1}\leq\Lambda} \mathbf{w}\cdot\sum_{i=1}^{m}\sum_{y\in[c]}\epsilon_{i,y}\boldsymbol{\Psi}(x_i,y)\right]$$

$$= \Lambda\mathbb{E}_{\boldsymbol{\epsilon}}\left[\left\|\sum_{i=1}^{m}\sum_{y\in[c]}\epsilon_{i,y}\boldsymbol{\Psi}(x_i,y)\right\|_{2,\infty}\right]$$

$$= \Lambda\mathbb{E}_{\boldsymbol{\epsilon}}\left[\max_y\left\|\sum_{i=1}^{m}\epsilon_{i,y}\Psi_y(x_i)\right\|_2\right]$$

$$\leq \Lambda\left(\mathbb{E}_{\boldsymbol{\epsilon}}\left[\max_y\left\|\sum_{i=1}^{m}\epsilon_{i,y}\Psi_y(x_i)\right\|_2^2\right]\right)^{1/2}$$

$$= \Lambda\left(\mathbb{E}_{\boldsymbol{\epsilon}}\left[\max_y\sum_{i=1}^{m}\left\|\Psi_y(x_i)\right\|_2^2+\sum_{i\neq j}\epsilon_{i,y}\epsilon_{j,y}\Psi_y(x_i)\cdot\Psi_y(x_j)\right]\right)^{1/2}$$

$$\leq \Lambda\left(\max_y\sum_{i=1}^{m}\left\|\Psi_y(x_i)\right\|_2^2+\mathbb{E}_{\boldsymbol{\epsilon}}\left[\max_y\sum_{i\neq j}\epsilon_{i,y}\epsilon_{j,y}\Psi_y(x_i)\cdot\Psi_y(x_j)\right]\right)^{1/2}$$

By Lemma 9 (or Massart's lemma), the following bound holds:

$$\mathbb{E}_{\boldsymbol{\epsilon}}\left[\max_y\sum_{i\neq j}\epsilon_{i,y}\epsilon_{j,y}\Psi_y(x_i)\cdot\Psi_y(x_j)\right] \leq mr_{2,\infty}\sqrt{\log(c)}.$$

Since, $\max_y\sum_{i=1}^{m}\|\Psi_y(x_i)\|_2^2 \leq mr_{2,\infty}^2$, we obtain that the following result holds:

$$\widehat{\mathfrak{R}}_S^G(\mathcal{H}_{2,1}) \leq \frac{\sqrt{2}\Lambda r_{2,\infty}(\log(c))^{1/4}}{\sqrt{m}},$$

and applying Theorem 1 completes the proof. $\qquad\square$

## A.8 VRM structured prediction learning bounds

Here, we give the proof of our structured prediction learning guarantees in the setting of Voted Risk Minimization. We will use the following lemma.

**Lemma 13.** *The function $\Phi^*$ is sub-additive: $\Phi^*(x+y) \leq \Phi^*(x) + \Phi^*(y)$, for all $x,y\in\mathbb{R}$.*

*Proof.* By the sub-additivity of the maximum function, for any $x,y\in\mathbb{R}$, the following upper bound holds for $\Phi^*(x+y)$:

$$\begin{aligned}
\Phi^*(x+y) = \min(M,\max(0,x+y)) &\leq \min(M,\max(0,x)+\max(0,y)) \\
&\leq \min(M,\max(0,x))+\min(M,\max(0,y)) \\
&= \Phi^*(x)+\Phi^*(y),
\end{aligned}$$

which completes the proof. $\qquad\square$

For the following proof, for any $\tau\geq 0$, the margin losses $R_{\rho,\tau}^{\mathrm{add}}(h)$ and $R_{\rho,\tau}^{\mathrm{mult}}(h)$ are defined as the population counterparts of the empirical losses define by (7) and (8).

**Theorem 3.** *Fix $\rho > 0$. For any $\delta > 0$, with probability at least $1-\delta$ over the draw of a sample $S$ of size $m$, each of the following inequalities holds for all $f\in\mathcal{F}$:*

$$R(f) - \widehat{R}_{S,\rho,1}^{add}(f) \leq \frac{4\sqrt{2}}{\rho}\sum_{t=1}^{T}\alpha_t\mathfrak{R}_m^G(H_{k_t}) + C(\rho,M,c,m,p),$$

$$R(f) - \widehat{R}_{S,\rho,1}^{mult}(f) \leq \frac{4\sqrt{2}M}{\rho}\sum_{t=1}^{T}\alpha_t\mathfrak{R}_m^G(H_{k_t}) + C(\rho,M,c,m,p).$$

*where*

$$C(\rho, M, c, m, p) = \frac{2M}{\rho}\sqrt{\frac{\log p}{m}} + 3M\sqrt{\left\lceil \frac{4}{\rho^2}\log\left(\frac{c^2\rho^2 m}{4\log p}\right)\right\rceil \frac{\log p}{m} + \frac{\log \frac{2}{\delta}}{2m}}.$$

*Proof.* The proof makes use of Theorem 1 and the proof techniques of Kuznetsov et al. [2014][Theorem 1] but requires a finer analysis both because of the general loss functions used here and because of the more complex structure of the hypothesis set.

For a fixed $\mathbf{h} = (h_1, \ldots, h_T)$, any $\boldsymbol{\alpha}$ in the probability simplex $\Delta$ defines a distribution over $\{h_1, \ldots, h_T\}$. Sampling from $\{h_1, \ldots, h_T\}$ according to $\boldsymbol{\alpha}$ and averaging leads to functions $g$ of the form $g = \frac{1}{n}\sum_{i=1}^T n_t h_t$ for some $\mathbf{n} = (n_1, \ldots, n_T) \in \mathbb{N}^T$, with $\sum_{t=1}^T n_t = n$, and $h_t \in \mathcal{H}_{k_t}$.

For any $\mathbf{N} = (N_1, \ldots, N_p)$ with $|\mathbf{N}| = n$, we consider the family of functions

$$G_{\mathcal{F},\mathbf{N}} = \left\{ \frac{1}{n}\sum_{k=1}^p \sum_{j=1}^{N_k} h_{k,j} \mid \forall (k,j) \in [p] \times [N_k], h_{k,j} \in H_k \right\},$$

and the union of all such families $G_{\mathcal{F},n} = \bigcup_{|\mathbf{N}|=n} G_{\mathcal{F},\mathbf{N}}$. Fix $\rho > 0$. For a fixed $\mathbf{N}$, the empirical factor graph Rademacher complexity of $G_{\mathcal{F},\mathbf{N}}$ can be bounded as follows for any $m \geq 1$:

$$\widehat{\mathfrak{R}}_S^G(G_{\mathcal{F},\mathbf{N}}) \leq \frac{1}{n}\sum_{k=1}^p N_k \widehat{\mathfrak{R}}_S^G(H_k),$$

which also implies the result for the true factor graph Rademacher complexities.

Thus, by Theorem 1, the following learning bound holds: for any $\delta > 0$, with probability at least $1 - \delta$, for all $g \in G_{\mathcal{F},\mathbf{N}}$,

$$R_{\rho,\frac{1}{2}}^{\mathrm{add}}(g) - \widehat{R}_{S,\rho,\frac{1}{2}}^{\mathrm{add}}(g) \leq \frac{1}{n}\frac{4\sqrt{2}}{\rho}\sum_{k=1}^p N_k \mathfrak{R}_m^G(H_k) + M\sqrt{\frac{\log\frac{1}{\delta}}{2m}}.$$

Since there are at most $p^n$ possible $p$-tuples $\mathbf{N}$ with $|\mathbf{N}| = n$,[3] by the union bound, for any $\delta > 0$, with probability at least $1 - \delta$, for all $g \in G_{\mathcal{F},n}$, we can write

$$R_{\rho,\frac{1}{2}}^{\mathrm{add}}(g) - \widehat{R}_{S,\rho,\frac{1}{2}}^{\mathrm{add}}(g) \leq \frac{1}{n}\frac{4\sqrt{2}}{\rho}\sum_{k=1}^p N_k \mathfrak{R}_m^G(H_k) + M\sqrt{\frac{\log\frac{p^n}{\delta}}{2m}}.$$

Thus, with probability at least $1 - \delta$, for all functions $g = \frac{1}{n}\sum_{i=1}^T n_t h_t$ with $h_t \in \mathcal{H}_{k_t}$, the following inequality holds

$$R_{\rho,\frac{1}{2}}^{\mathrm{add}}(g) - \widehat{R}_{S,\rho,\frac{1}{2}}^{\mathrm{add}}(g) \leq \frac{1}{n}\frac{4\sqrt{2}}{\rho}\sum_{k=1}^p \sum_{t:k_t=k} n_t \mathfrak{R}_m^G(H_{k_t}) + M\sqrt{\frac{\log\frac{p^n}{\delta}}{2m}}.$$

Taking the expectation with respect to $\boldsymbol{\alpha}$ and using $\mathbb{E}_{\boldsymbol{\alpha}}[n_t/n] = \alpha_t$, we obtain that for any $\delta > 0$, with probability at least $1 - \delta$, for all $g$, we can write

$$\mathbb{E}_{\boldsymbol{\alpha}}[R_{\rho,\frac{1}{2}}^{\mathrm{add}}(g) - \widehat{R}_{S,\rho,\frac{1}{2}}^{\mathrm{add}}(g)] \leq \frac{4\sqrt{2}}{\rho}\sum_{t=1}^T \alpha_t \mathfrak{R}_m^G(H_{k_t}) + M\sqrt{\frac{\log\frac{p^n}{\delta}}{2m}}.$$

Fix $n \geq 1$. Then, for any $\delta_n > 0$, with probability at least $1 - \delta_n$,

$$\mathbb{E}_{\boldsymbol{\alpha}}[R_{\rho,\frac{1}{2}}^{\mathrm{add}}(g) - \widehat{R}_{S,\rho,\frac{1}{2}}^{\mathrm{add}}(g)] \leq \frac{4\sqrt{2}}{\rho}\sum_{t=1}^T \alpha_t \mathfrak{R}_m^G(H_{k_t}) + M\sqrt{\frac{\log\frac{p^n}{\delta_n}}{2m}}.$$

Choose $\delta_n = \frac{\delta}{2p^{n-1}}$ for some $\delta > 0$, then for $p \geq 2$, $\sum_{n \geq 1} \delta_n = \frac{\delta}{2(1-1/p)} \leq \delta$. Thus, for any $\delta > 0$ and any $n \geq 1$, with probability at least $1 - \delta$, the following holds for all $g$:

$$\mathbb{E}_{\boldsymbol{\alpha}}[R^{\text{add}}_{\rho, \frac{1}{2}}(g) - \widehat{R}^{\text{add}}_{S, \rho, \frac{1}{2}}(g)] \leq \frac{4\sqrt{2}}{\rho} \sum_{t=1}^{T} \alpha_t \mathfrak{R}^G_m(H_{k_t}) + M\sqrt{\frac{\log \frac{2p^{2n-1}}{\delta}}{2m}}. \tag{13}$$

Now, for any $f = \sum_{t=1}^{T} \alpha_t h_t \in \mathcal{F}$ and any $g = \frac{1}{n} \sum_{i=1}^{T} n_t h_t$, using (4), we can upper bound $R(f)$, the generalization error of $f$, as follows:

$$R(f) = \mathbb{E}\left[ L(\mathsf{f}(x), y) 1_{\rho_f(x,y) \leq 0} \right] \tag{14}$$

$$\leq \mathbb{E}\left[ L(\mathsf{f}(x), y) 1_{\rho_f(x,y)-(g(x,y)-g(x,y_f)) < -\rho/2} \right] + \mathbb{E}\left[ L(\mathsf{f}(x), y) 1_{g(x,y)-g(x,y_f) \leq \rho/2} \right]$$

$$\leq M \Pr\left[ \rho_f(x,y) - (g(x,y) - g(x,y_f)) < -\rho/2 \right] + \mathbb{E}\left[ L(\mathsf{f}(x), y) 1_{g(x,y)-g(x,y_f) \leq \rho/2} \right],$$

where for any function $\varphi \colon \mathcal{X} \times \mathcal{Y} \to [0,1]$, we define $y_\varphi$ as follows: $y_\varphi = \operatorname{argmax}_{y' \neq y} \varphi(x,y)$. Using the same arguments as in the proof of Lemma 4, one can show that

$$\mathbb{E}\left[ L(\mathsf{f}(x), y)) 1_{g(x,y)-g(x,y_f) < \rho/2} \right] \leq R^{\text{add}}_{\rho, \frac{1}{2}}(g).$$

We now give a lower-bound on $\widehat{R}^{\text{add}}_{S, \rho, 1}(f)$ in terms of $R^{\text{add}}_{S, \rho, \frac{1}{2}}(g)$. To do so, we start with the expression of $\widehat{R}^{\text{add}}_{S, \rho, \frac{1}{2}}(g)$:

$$\widehat{R}^{\text{add}}_{S, \rho, \frac{1}{2}}(g) = \mathbb{E}_{(x,y) \sim S}\left[ \Phi^*\left( \max_{y' \neq y} \mathsf{L}(y', y) + \tfrac{1}{2} - \tfrac{1}{\rho}[g(x,y) - g(x,y')] \right) \right]$$

By the sub-additivity of $\max$, we can write

$$\max_{y' \neq y} \mathsf{L}(y', y) + \tfrac{1}{2} - \tfrac{1}{\rho}[g(x,y) - g(x,y')]$$

$$\leq \max_{y' \neq y}\left\{ L(y, y') + 1 - \frac{f(x,y) - f(x,y')}{\rho} \right\}$$

$$+ \max_{y' \neq y}\left\{ -\frac{1}{2} + \frac{f(x,y) - f(x,y')}{\rho} - \frac{g(x,y) - g(x,y')}{\rho} \right\} = X + Y,$$

where $X$ and $Y$ are defined by

$$X = \max_{y' \neq y}\left( L(y, y') + 1 - \frac{f(x,y) - f(x,y')}{\rho} \right),$$

$$Y = -\frac{1}{2} + \max_{y' \neq y}\left( \frac{f(x,y) - f(x,y')}{\rho} - \frac{g(x,y) - g(x,y')}{\rho} \right).$$

In view of that, since $\Phi^*$ is non-decreasing and sub-additive (Lemma 13), we can write

$$\widehat{R}^{\text{add}}_{S, \rho, \frac{1}{2}}(g) \leq \mathbb{E}_{(x,y) \sim S}[\Phi^*(X + Y)] \tag{15}$$

$$\leq \mathbb{E}_{(x,y) \sim S}[\Phi^*(X) + \Phi^*(Y)] = \mathbb{E}_{(x,y) \sim S}[\Phi^*(X)] + \mathbb{E}_{(x,y) \sim S}[\Phi^*(Y)]$$

$$= \widehat{R}^{\text{add}}_{S, \rho, 1}(f) + \mathbb{E}_{(x,y) \sim S}[\Phi^*(Y)]$$

$$\leq \widehat{R}^{\text{add}}_{S, \rho, 1}(f) + M \mathbb{E}_{(x,y) \sim S}[1_{Y > 0}]$$

$$= \widehat{R}^{\text{add}}_{S, \rho, 1}(f) + M \Pr_{(x,y) \sim S}\left[ \max_{y' \neq y}\{ f(x,y) - g(x,y) + (g(x,y') - f(x,y')) \} > \rho/2 \right].$$

Combining (14) and (15) shows that $R(f) - \widehat{R}_{S,\rho,1}^{\text{add}}(f)$ is bounded by

$$R_{\rho,\frac{1}{2}}^{\text{add}}(g) - \widehat{R}_{S,\rho,\frac{1}{2}}^{\text{add}}(g) + M \Pr\left[\rho_f(x,y) - (g(x,y) - g(x,y_f)) < -\rho/2\right]$$
$$+ M \Pr_{(x,y)\sim S}\left[\max_{y'\neq y}\{f(x,y) - g(x,y) + (g(x,y') - f(x,y'))\} > \rho/2\right].$$

Taking the expectation with respect to $\boldsymbol{\alpha}$ shows that $R(f) - \widehat{R}_{S,\rho,1}^{\text{add}}(f)$ is bounded by

$$\mathbb{E}_{\boldsymbol{\alpha}}\left[R_{\rho,\frac{1}{2}}^{\text{add}}(g) - \widehat{R}_{S,\rho,\frac{1}{2}}^{\text{add}}(g)\right] + M \mathbb{E}_{(x,y)\sim\mathcal{D},\boldsymbol{\alpha}}\left[1_{\rho_f(x,y)-(g(x,y)-g(x,y_f))<-\rho/2}\right]$$
$$+ M \mathbb{E}_{(x,y)\sim S,\boldsymbol{\alpha}}\left[1_{\max_{y'\neq y}\{f(x,y)-g(x,y)+(g(x,y')-f(x,y'))\}>\rho/2}\right]. \quad (16)$$

By Hoeffding's bound, the following holds:

$$\mathbb{E}_{\boldsymbol{\alpha}}\left[1_{\rho_f(x,y)-(g(x,y)-g(x,y_f))<-\rho/2}\right] = \Pr_{\boldsymbol{\alpha}}\left[(f(x,y) - f(x,y_f)) - (g(x,y) - g(x,y_f)) < -\rho/2\right]$$
$$\leq e^{-n\rho^2/8}.$$

Similarly, using the union bound and Hoeffding's bound, the third expectation term appearing in (16) can be bounded as follows:

$$\mathbb{E}_{\boldsymbol{\alpha}}\left[1_{\max_{y'\neq y}\{f(x,y)-g(x,y)+(g(x,y')-f(x,y'))\}>\rho/2}\right]$$
$$= \Pr_{\boldsymbol{\alpha}}\left[\max_{y'\neq y}\{f(x,y) - g(x,y) + (g(x,y') - f(x,y'))\} > \rho/2\right]$$
$$\leq \sum_{y'\neq y} \Pr_{\boldsymbol{\alpha}}\left[f(x,y) - g(x,y) + (g(x,y') - f(x,y')) > \rho/2\right]$$
$$\leq (c-1)e^{-n\rho^2/8}.$$

Thus, for any fixed $f$, we can write

$$R(f) - \widehat{R}_{S,\rho,1}^{\text{add}}(f) \leq cMe^{-n\rho^2/8} + \mathbb{E}_{\boldsymbol{\alpha}}\left[R_{\rho,\frac{1}{2}}^{\text{add}}(g) - \widehat{R}_{S,\rho,\frac{1}{2}}^{\text{add}}(g)\right].$$

Therefore, the following quantity upper bounds $\sup_f R(f) - \widehat{R}_{S,\rho,1}^{\text{add}}(f)$:

$$cMe^{-n\rho^2/8} + \sup_g \mathbb{E}_{\boldsymbol{\alpha}}\left[R_{\rho,\frac{1}{2}}^{\text{add}}(g) - \widehat{R}_{S,\rho,\frac{1}{2}}^{\text{add}}(g)\right],$$

and, in view of (13), for any $\delta > 0$ and any $n \geq 1$, with probability at least $1 - \delta$, the following holds for all $f$:

$$R(f) - \widehat{R}_{S,\rho,1}^{\text{add}}(f) \leq cMe^{-n\rho^2/8} + \frac{4\sqrt{2}}{\rho}\sum_{t=1}^{T}\alpha_t\mathfrak{R}_m^G(H_{k_t}) + M\sqrt{\frac{\log\frac{2p^{2n-1}}{\delta}}{2m}}.$$

Choosing $n = \left\lceil \frac{4}{\rho^2}\log\left(\frac{c^2\rho^2 m}{4\log p}\right)\right\rceil$ yields the following inequality:[4]

$$R(f) - \widehat{R}_{S,\rho,1}^{\text{add}}(f) \leq \frac{4\sqrt{2}}{\rho}\sum_{t=1}^{T}\alpha_t\mathfrak{R}_m^G(H_{k_t}) + \frac{2M}{\rho}\sqrt{\frac{\log p}{m}}$$
$$+ 3M\sqrt{\left\lceil\frac{4}{\rho^2}\log\left(\frac{c^2\rho^2 m}{4\log p}\right)\right\rceil\frac{\log p}{m} + \frac{\log\frac{2}{\delta}}{2m}},$$

and concludes the proof. $\qquad\qquad\square$

Table 1: Description of datasets.

| Dataset | Full name | Sentences | Tokens | Unique tokens | Labels |
|---|---|---|---|---|---|
| Basque | Basque UD Treebank | 8993 | 121443 | 26679 | 16 |
| Chinese | Chinese Treebank 6.0 | 28295 | 782901 | 47570 | 37 |
| Dutch | UD Dutch Treebank | 13735 | 200654 | 29123 | 16 |
| English | UD English Web Treebank | 16622 | 254830 | 23016 | 17 |
| Finnish | Finnish UD Treebank | 13581 | 181018 | 53104 | 12 |
| Finnish-FTB | UD_Finnish-FTB | 18792 | 160127 | 46756 | 15 |
| Hindi | UD Hindi Treebank | 16647 | 351704 | 19232 | 16 |
| Tamil | UD Tamil Treebank | 600 | 9581 | 3583 | 14 |
| Turkish | METU-Sabanci Turkish Treebank | 5635 | 67803 | 19125 | 32 |
| Twitter | Tweebank | 929 | 12318 | 4479 | 25 |

By applying Theorem 7 instead of Theorem 1 and keeping track of the slightly increased exponential concentration terms in the proof above, we arrive at the following analogue of Theorem 3 in terms of empirical complexities:

**Theorem 14.** *Fix $\rho > 0$. For any $\delta > 0$, with probability at least $1 - \delta$ over the draw of a sample $S$ of size $m$, each of the following inequalities holds for all $f \in \mathcal{F}$:*

$$R(f) - \widehat{R}_{S,\rho,1}^{add}(f) \leq \frac{4\sqrt{2}}{\rho} \sum_{t=1}^{T} \alpha_t \hat{\mathfrak{R}}_m^G(H_{k_t}) + C(\rho, M, c, m, p),$$

$$R(f) - \widehat{R}_{S,\rho,1}^{mult}(f) \leq \frac{4\sqrt{2}M}{\rho} \sum_{t=1}^{T} \alpha_t \hat{\mathfrak{R}}_m^G(H_{k_t}) + C(\rho, M, c, m, p).$$

*where*

$$C(\rho, M, c, m, p) = \frac{2M}{\rho} \sqrt{\frac{\log p}{m}} + 9M \sqrt{\left\lceil \frac{4}{\rho^2} \log \left( \frac{c^2 \rho^2 m}{4 \log p} \right) \right\rceil \frac{\log p}{m} + \frac{\log \frac{2}{\delta}}{2m}}.$$

### A.9 General upper bound on the loss based on convex surrogates

Here, we present the proof of a general upper bound on a loss function in terms of convex surrogates.

**Lemma 4.** *For any $u \in \mathbb{R}_+$, let $\Phi_u \colon \mathbb{R} \to \mathbb{R}$ be an upper bound on $v \mapsto u\mathbf{1}_{v \leq 0}$. Then, the following upper bound holds for any $h \in \mathcal{H}$ and $(x, y) \in \mathcal{X} \times \mathcal{Y}$,*

$$\mathsf{L}(\mathsf{h}(x), y) \leq \max_{y' \neq y} \Phi_{\mathsf{L}(y', y)}(h(x, y) - h(x, y')). \tag{17}$$

*Proof.* If $\mathsf{h}(x) = y$, then $\mathsf{L}(\mathsf{h}(x), y) = 0$ and the result follows. Otherwise, $\mathsf{h}(x) \neq y$ and the following bound holds:

$$\begin{aligned}
\mathsf{L}(\mathsf{h}(x), y) &= \mathsf{L}(\mathsf{h}(x), y)\mathbf{1}_{\rho_h(x,y) \leq 0} \\
&\leq \Phi_{\mathsf{L}(\mathsf{h}(x),y)}(\rho_h(x, y)) \\
&= \Phi_{\mathsf{L}(\mathsf{h}(x),y)}(h(x, y) - \max_{y' \neq y} h(x, y')) \\
&= \Phi_{\mathsf{L}(\mathsf{h}(x),y)}(h(x, y) - h(x, \mathsf{h}(x))) \\
&\leq \max_{y' \neq y} \Phi_{\mathsf{L}(y',y)}(h(x, y) - h(x, y')),
\end{aligned}$$

which concludes the proof. $\qquad\square$

## B Experiments

### B.1 Datasets

This section reports the results of preliminary experiments with the VCRF algorithm. The experiments in this section are meant to serve as a proof of concept of the benefits of VRM-type regularization as

suggested by the theory developed in this paper. We leave an extensive experimental study of other aspects of our theory, including general loss functions, convex surrogates and $p$-norms, to future work.

For our experiments, we chose the part-of-speech task (POS) that consists of labeling each word of a sentence with its correct part-of-speech tag. We used 10 POS datasets: `Basque`, `Chinese`, `Dutch`, `English`, `Finnish`, `Finnish-FTB`, `Hindi`, `Tamil`, `Turkish` and `Twitter`. The detailed description of these datasets is in Appendix B.1. Our VCRF algorithm can be applied with a variety of different families of feature functions $H_k$ mapping $\mathcal{X} \times \mathcal{Y}$ to $\mathbb{R}$. Details concerning features and complexity penalties $r_k$s are provided in Appendix B.2, while an outline of our hyperparameter selection and cross-validation procedure is given in Appendix B.3.

The average error and the standard deviation of the errors are reported in Table 2 for each data set. Our results show that VCRF provides a statistically significant improvement over $L_1$-CRF on every dataset, with the exception of `English` and `Dutch`. One-sided paired $t$-test at $5\%$ level was used to assess the significance of the results. It should be noted that for all of the significant results, VCRF outperformed $L_1$-CRF on every fold. Furthermore, our results indicate that VCRF tends to produce models that are sparser than those of $L_1$-CRF. This is highlighted in Table 3 of Appendix B.2. As can be seen, VCRF tends to produce models that are much more sparse due to its heavy penalization on the large number of higher-order features. In a separate set of experiments, we have also tested the robustness of our algorithm to erroneous annotations and noise. The details and the results of these experiments are given in Appendix B.4.

Further details on the datasets and the specific features as well as more experimental results are provided below.

Table 1 provides some statistics for each of the datasets that we use. These datasets span a variety of sizes, in terms of sentence count, token count, and unique token count. Most are annotated under the Universal Dependencies (UD) annotation system, with the exception of the Chinese (Palmer et al. [2007]), Turkish (Oflazer et al. [2003], Atalay et al. [2003]), and Twitter (Gimpel et al. [2011], Owoputi et al. [2013]) datasets.

## B.2  Features and complexities

The standard features that are used in POS tagging are usually binary indicators that signal the occurrence of certain words, tags or other linguistic constructs such as suffixes, prefixes, punctuation, capitalization or numbers in a window around a given position in the sequence. In our experiments, we use the union of a broad family of products of such indicator functions. Let $V$ denote the input vocabulary over alphabet $\Sigma$. For $x \in V$ and $t \geq 0$, let $\mathrm{suff}(x, t)$ be the suffix of length $t$ for the word $x$ and $\mathrm{pref}(x, t)$ the prefix. Then for $k_1, k_2, k_3 \geq 0$, we can define the following three families of base features:

$$H_{k_1}^{\mathrm{w}}(s) = \left\{ x \mapsto \mathbf{1}_{x_{s-t+1}^{s+r} = x'} : t, r \in \mathbb{N}, r + t = k_1, x' \in V^{k_1} \right\},$$
$$H_{k_2}^{\mathrm{tag}}(s) = \left\{ y \mapsto \mathbf{1}_{y_{s-k_2+1}^{s} = y'} : y' \in \Delta^{k_2} \right\},$$
$$H_{k_3}^{\mathrm{sp}}(s) = \left\{ x \mapsto \mathbf{1}_{\mathrm{suff}(x_s,t)=S} \mathbf{1}_{\mathrm{pref}(x_s,r)=P} : t, r \in \mathbb{N}, t + r = k_3, S \in \Sigma^t, P \in \Sigma^r \right\}.$$

We can then define a family of features $H_{k_1,k_2,k_3}$ that consists of functions of the form

$$\Psi(x, y) = \sum_{s=1}^{l} \psi(x, y, s),$$

where $\psi(x, y, s) = h_1(x) h_2(y) h_3(x)$, for some $h_1 \in \mathcal{H}_{k_1}^{\mathrm{w}}(s), h_2 \in \mathcal{H}_{k_2}^{\mathrm{tag}}(s), h_3 \in \mathcal{H}_{k_3}^{\mathrm{sp}}(s)$.

As an example, consider the following sentence:

| DET | NN | VBD | RB | JJ |
|-----|-----|-----|-----|-----|
| The | cat | was | surprisingly | agile |

Figure 2: Example of features for a POS task.

Table 2: Experimental results for both VCRF and CRF. VCRF refers to the conditional random field objective with both VRM-style regularization and $L_1$ regularization while CRF refers to the objective with only $L_1$ regularization. Boldfaced results are statistically significant at a 5% confidence level.

| | VCRF error (%) | | CRF error(%) | |
|---|---|---|---|---|
| Dataset | Token | Sentence | Token | Sentence |
| Basque | **7.26 ± 0.13** | **57.67 ± 0.82** | 7.68 ± 0.20 | 59.78 ± 1.39 |
| Chinese | **7.38 ± 0.15** | **67.73 ± 0.46** | 7.67 ± 0.12 | 68.88 ± 0.49 |
| Dutch | 5.97 ± 0.08 | 49.27 ± 0.71 | 6.01 ± 0.92 | 49.48 ± 1.02 |
| English | 5.51 ± 0.04 | 44.40 ± 1.30 | 5.51 ± 0.06 | 44.32 ± 1.31 |
| Finnish | **7.48 ± 0.05** | **55.96 ± 0.64** | 7.86 ± 0.13 | 57.17 ± 1.36 |
| Finnish-FTB | **9.79 ± 0.22** | **51.23 ± 1.21** | 10.55 ± 0.22 | 52.98 ± 0.75 |
| Hindi | **4.84 ± 0.10** | **51.69 ± 1.07** | 4.93 ± 0.08 | 53.18 ± 0.75 |
| Tamil | **19.82 ± 0.69** | **89.83 ± 2.13** | 22.50 ± 1.57 | 92.00 ± 1.54 |
| Turkish | **11.28 ± 0.40** | **59.63 ± 1.55** | 11.69 ± 0.37 | 61.15 ± 1.01 |
| Twitter | **17.98 ± 1.25** | **75.57 ± 1.25** | 19.81 ± 1.09 | 76.96 ± 1.37 |

Then, at position $s = 3$, the following features $h_1 \in \mathcal{H}_3^{\mathrm{w}}(3)$, $h_2 \in \mathcal{H}_2^{\mathrm{tag}}(3)$, $h_3 \in \mathcal{H}_1^{\mathrm{sp}}(3)$ would activate:

$$h_1(x) = \mathbf{1}_{x_2=\text{'was'},\, x_3=\text{'surprisingly'},\, x_4=\text{'agile'}}(x)$$
$$h_2(y) = \mathbf{1}_{y_2=\text{'VBD'},\, y_3=\text{'RB'}}(y)$$
$$h_3(x) = \mathbf{1}_{\mathrm{suff}(x_3,2)=\text{'ly'}}(x).$$

See Figure 2 for an illustration.

Now, recall that the VCRF algorithm requires knowledge of complexities $r(H_{k_1,k_2,k_3})$. By definition of the hypothesis set and $r_k$s

$$r(H_{k_1,k_2,k_3}) \leq \sqrt{\frac{2(k_1 \log |V| + k_2 \log |\Delta| + k_3 \log |\Sigma|)}{m}}, \tag{18}$$

which is precisely the complexity penalty used in our experiments.

The impact of this added penalization can be seen in Table 3, where it is seen that the number of non-zero features for VCRF can be dramatically smaller than the number for $L_1$-regularized CRF.

### B.3 Hyperparameter tuning and cross-validation

Recall that the VCRF algorithm admits two hyperparameters $\lambda$ and $\beta$. In our experiments, we optimized over $\lambda, \beta \in \{1, 0.5, 10^{-1}, \ldots, 10^{-5}, 0\}$. We compared VCRF against $L_1$-regularized CRF, which is the special case of VCRF with $\lambda = 0$. For gradient computation, we used the procedure in Section D.2.1, which is agnostic to the choice of the underlying loss function. While our algorithms can be used with very general families of loss functions this choice allows an easy direct comparison with the CRF algorithm. We ran each algorithm for 50 full passes over the entire training set or until convergence.

In each of the experiments, we used 5-fold cross-validation for model selection and performance evaluation. Each dataset was randomly partitioned into 5 folds, and each algorithm was run 5 times, with a different assignment of folds to the training set, validation set and test set for each run. For each run $i \in \{0, \ldots, 4\}$, fold $i$ was used for validation, fold $i + 1 (\bmod\ 5)$ was used for testing, and the remaining folds were used for training. In each run, we selected the parameters that had the lowest token error on the validation set and then measured the token and sentence error of those parameters on the test set. The average error and the standard deviation of the errors are reported in Table 2 for each data set.

Table 3: Average number of features for VCRF and $L_1$-CRF.

| Dataset | VCRF | CRF | Ratio |
|---|---|---|---|
| Basque | 7028 | 94712653 | 0.00007 |
| Chinese | 219736 | 552918817 | 0.00040 |
| Dutch | 2646231 | 2646231 | 1.00000 |
| English | 4378177 | 357011992 | 0.01226 |
| Finnish | 32316 | 89333413 | 0.00036 |
| Finnish-FTB | 53337 | 5735210 | 0.00930 |
| Hindi | 108800 | 448714379 | 0.00024 |
| Tamil | 1583 | 668545 | 0.00237 |
| Turkish | 498796 | 3314941 | 0.15047 |
| Twitter | 18371 | 26660216 | 0.000689 |

Table 4: Experimental results of both VCRF and CRF with 20% random noise added to the training set. Labels of tokens are flipped uniformly at random with 20% probability. Boldfaced results are statistically significant at a 5% confidence level.

| | VCRF error (%) | | CRF error(%) | |
|---|---|---|---|---|
| Dataset | Token | Sentence | Token | Sentence |
| Basque | **9.13 ± 0.18** | **67.43 ± 0.93** | 9.42 ± 0.31 | 68.61 ± 1.08 |
| Chinese | **96.43 ± 0.33** | **100.00 ± 0.01** | 96.81 ± 0.43 | 100.00 ± 0.01 |
| Dutch | **8.16 ± 0.52** | **62.15 ± 1.77** | 8.57 ± 0.30 | 63.55 ± 0.87 |
| English | **8.79 ± 0.23** | **61.27 ± 1.21** | 9.20 ± 0.11 | 63.60 ± 1.18 |
| Finnish | **9.38 ± 0.27** | **64.96 ± 0.89** | 9.62 ± 0.18 | 65.91 ± 0.93 |
| Finnish-FTB | **11.39 ± 0.29** | **72.56 ± 1.30** | 11.76 ± 0.25 | 73.63 ± 1.19 |
| Hindi | **6.63 ± 0.51** | **63.84 ± 2.86** | 7.85 ± 0.33 | 71.93 ± 1.20 |
| Tamil | 20.77 ± 0.70 | 93.00 ± 1.35 | 21.36 ± 0.86 | 93.50 ± 1.78 |
| Turkish | 14.28 ± 0.46 | 69.72 ± 1.51 | 14.31 ± 0.53 | 69.62 ± 2.04 |
| Twitter | 90.92 ± 1.67 | 100.00 ± 0.00 | 92.27 ± 0.71 | 100.00 ± 0.00 |

### B.4 More experiments

In this section, we present our results for a POS tagging task when noise is artificially injected into the labels. Specifically, for tokens corresponding to features that commonly appear in the dataset (at least five times in our experiments), we flip their associated POS label to some other arbitrary label with 20% probability.

The results of these experiments are given in Table 4. They demonstrate that VCRF outperforms $L_1$-CRF in the majority of cases. Moreover, these differences can be magnified from the original scenario, as can be seen on the English and Twitter datasets.

## C Voted Structured Boosting (VStructBoost)

In this section, we consider algorithms based on the StructBoost surrogate loss, where we choose $\Phi_u(v) = ue^{-v}$. Let $\delta\Psi(x, y, y') = \Psi(x, y) - \Psi(x, y')$. This then leads to the following optimization problem:

$$\min_{\mathbf{w}} \frac{1}{m} \sum_{i=1}^{m} \max_{y \neq y_i} \mathsf{L}(y, y_i) e^{-\mathbf{w} \cdot \delta\Psi(x_i, y_i, y)} + \sum_{k=1}^{p} (\lambda r_k + \beta) \|\mathbf{w}_k\|_1. \quad (19)$$

One disadvantage of this formulation is that the first term of the objective is not differentiable. Upper bounding the maximum by a sum leads to the following optimization problem:

$$\min_{\mathbf{w}} \frac{1}{m} \sum_{i=1}^{m} \sum_{y \neq y_i} \mathsf{L}(y, y_i) e^{-\mathbf{w} \cdot \delta\Psi(x_i, y_i, y)} + \sum_{k=1}^{p} (\lambda r_k + \beta) \|\mathbf{w}_k\|_1. \quad (20)$$

We refer to the learning algorithm based on the optimization problem (20) as VStructBoost. To the best of our knowledge, the formulations (19) and (20) are new, even with the standard $L_1$- or $L_2$-regularization.

## D Optimization solutions

Here, we show how the optimization problems in (10) and (20) can be solved efficiently when the feature vectors admit a particular factor graph decomposition that we refer to as Markov property.

### D.1 Markovian features

We will consider in what follows the common case where $\mathcal{Y}$ is a set of sequences of length $l$ over a finite alphabet $\Delta$ of size $r$. Other structured problems can be treated in similar ways. We will denote by $\varepsilon$ the empty string and for any sequence $y = (y_1, \ldots, y_l) \in \mathcal{Y}$, we will denote by $y_s^{s'} = (y_s, \ldots, y_{s'})$ the substring of $y$ starting at index $s$ and ending at $s'$. For convenience, for $s \leq 0$, we define $y_s$ by $y_s = \varepsilon$.

One common assumption that we shall adopt here is that the feature vector $\boldsymbol{\Psi}$ admits a *Markovian property of order* $q$. By this, we mean that it can be decomposed as follows for any $(x, y) \in \mathcal{X} \times \mathcal{Y}$:

$$\boldsymbol{\Psi}(x, y) = \sum_{s=1}^{l} \boldsymbol{\psi}(x, y_{s-q+1}^s, s). \tag{21}$$

for some position-dependent feature vector function $\boldsymbol{\psi}$ defined over $\mathcal{X} \times \Delta^q \times [l]$. This also suggests a natural decomposition of the family of feature vectors $\boldsymbol{\Psi} = (\boldsymbol{\Psi}_1, \ldots, \boldsymbol{\Psi}_p)$ for the application of VRM principle where $\boldsymbol{\Psi}_k$ is a Markovian feature vector of order $k$. Thus, $\mathcal{F}_k$ then consists of the family of Markovian feature functions of order $k$. We note that we can write $\boldsymbol{\Psi} = \sum_{k=1}^{p} \tilde{\boldsymbol{\Psi}}_k$ with $\tilde{\boldsymbol{\Psi}}_k = (0, \ldots, \boldsymbol{\Psi}_k, \ldots, 0)$. In the following, abusing the notation, we will simply write $\boldsymbol{\Psi}_k$ instead of $\tilde{\boldsymbol{\Psi}}_k$. Thus, for any $x \in \mathcal{X}$ and $y \in \mathcal{Y}$,[5]

$$\boldsymbol{\Psi}(\mathbf{x}, y) = \sum_{k=1}^{p} \boldsymbol{\Psi}_k(x, y). \tag{22}$$

For any $k \in [1, p]$, let $\boldsymbol{\psi}_k$ denote the position-dependent feature vector function corresponding to $\boldsymbol{\Psi}_k$. Also, for any $x \in \mathcal{X}$ and $y \in \Delta^l$, define $\widetilde{\boldsymbol{\psi}}$ by $\widetilde{\boldsymbol{\psi}}(x, y_{s-p+1}^s, s) = \sum_{k=1}^{p} \boldsymbol{\psi}_k(x, y_{s-k+1}^s, s)$. Observe then that we can write

$$\begin{aligned}
\boldsymbol{\Psi}(x, y) = \sum_{k=1}^{p} \boldsymbol{\Psi}_k(x, y) &= \sum_{k=1}^{p} \sum_{s=1}^{l} \boldsymbol{\psi}_k(x, y_{s-k+1}^s, s) \\
&= \sum_{s=1}^{l} \sum_{k=1}^{p} \boldsymbol{\psi}_k(x, y_{s-k+1}^s, s) \\
&= \sum_{s=1}^{l} \widetilde{\boldsymbol{\psi}}(x_i, y_{s-p+1}^s, s).
\end{aligned} \tag{23}$$

In Sections D.2 and D.3, we describe algorithms for efficiently computing the gradient by leveraging the underlying graph structure of the problem.

### D.2 Efficient gradient computation for VCRF

In this section, we show how Gradient Descent (GD) and Stochastic Gradient Descent (SGD) can be used to solve the optimization problem of VCRF. To do so, we will show how the subgradient of the contribution to the objective function of a given point $x_i$ can be computed efficiently. Since the computation of the subgradient of the regularization term presents no difficulty, it suffices to show that the gradient of $F_i$, the contribution of point $x_i$ to the empirical loss term for an arbitrary $i \in [m]$, can be computed efficiently. In the special case of the Hamming loss or when loss is omitted from the objective altogether, this coincides with the standard CRF training procedure. We extend this to more general families of loss function.

Fix $i \in [m]$. For the VCRF objective, $F_i$ can be rewritten as follows:

$$F_i(\mathbf{w}) = \frac{1}{m} \log \left( \sum_{y \in \mathcal{Y}} e^{\mathsf{L}(y, y_i) - \mathbf{w} \cdot \delta \boldsymbol{\Psi}(x_i, y_i, y)} \right) = \frac{1}{m} \log \left( \sum_{y \in \mathcal{Y}} e^{\mathsf{L}(y, y_i) + \mathbf{w} \cdot \boldsymbol{\Psi}(x_i, y)} \right) - \frac{\mathbf{w} \cdot \boldsymbol{\Psi}(x_i, y_i)}{m}.$$

The following lemma gives the expression of the gradient of $F_i$ and helps identify the key computationally challenging terms $\mathsf{q}_{\mathbf{w}}$.

**Lemma 15.** *The gradient of $F_i$ at any $\mathbf{w}$ can be expressed as follows:*

$$\nabla F_i(\mathbf{w}) = \frac{1}{m} \sum_{s=1}^{l} \sum_{\mathbf{z} \in \Delta^p} \left[ \sum_{y: \, y_{s-p+1}^s = \mathbf{z}} \mathsf{q}_{\mathbf{w}}(y) \right] \widetilde{\psi}(x_i, \mathbf{z}, s) - \frac{\Psi(x_i, y_i)}{m},$$

*where, for all $y \in \mathcal{Y}$,*

$$\mathsf{q}_{\mathbf{w}}(y) = \frac{e^{\mathsf{L}(y, y_i) + \mathbf{w} \cdot \Psi(x_i, y)}}{Z_{\mathbf{w}}},$$

$$Z_{\mathbf{w}} = \sum_{y \in \mathcal{Y}} e^{\mathsf{L}(y, y_i) + \mathbf{w} \cdot \Psi(x_i, y)}.$$

*Proof.* In view of the expression of $F_i$ given above, the gradient of $F_i$ at any $\mathbf{w}$ is given by

$$\nabla F_i(\mathbf{w}) = \frac{1}{m} \sum_{y \in \mathcal{Y}} \frac{e^{\mathsf{L}(y, y_i) + \mathbf{w} \cdot \Psi(x_i, y)}}{\sum_{\tilde{y} \in \mathcal{Y}} e^{\mathsf{L}(\tilde{y}, y_i) + \mathbf{w} \cdot \Psi(x_i, \tilde{y})}} \Psi(x_i, y) - \frac{\Psi(x_i, y_i)}{m}$$

$$= \frac{1}{m} \mathbb{E}_{y \sim \mathsf{q}_{\mathbf{w}}} [\Psi(x_i, y)] - \frac{\Psi(x_i, y_i)}{m}.$$

By (23), we can write

$$\mathbb{E}_{y \sim \mathsf{q}_{\mathbf{w}}} [\Psi(x_i, y)] = \sum_{y \in \Delta^l} \mathsf{q}_{\mathbf{w}}(y) \sum_{s=1}^{l} \widetilde{\psi}(x_i, y_{s-p+1}^s, s) = \sum_{s=1}^{l} \sum_{\mathbf{z} \in \Delta^p} \left[ \sum_{y: \, y_{s-p+1}^s = \mathbf{z}} \mathsf{q}_{\mathbf{w}}(y) \right] \widetilde{\psi}(x_i, \mathbf{z}, s),$$

which completes the proof. $\qquad\qquad\qquad\qquad\qquad\qquad\qquad\qquad\qquad\qquad\qquad\square$

The lemma implies that the key computation in the gradient is

$$\mathsf{Q}_{\mathbf{w}}(\mathbf{z}, s) = \sum_{y: \, y_{s-p+1}^s = \mathbf{z}} \mathsf{q}_{\mathbf{w}}(y) = \sum_{y: \, y_{s-p+1}^s = \mathbf{z}} \frac{e^{\mathsf{L}(y, y_i)} \prod_{t=1}^{l} e^{\mathbf{w} \cdot \widetilde{\psi}(x_i, y_{t-p+1}^t, t)}}{Z_{\mathbf{w}}}, \tag{24}$$

for all $s \in [l]$ and $\mathbf{z} \in \Delta^p$. The sum defining these terms is over a number of sequences $y$ that is exponential in $|\Delta|$. However, we will show in the following sections how to efficiently compute $\mathsf{Q}_{\mathbf{w}}(\mathbf{z}, s)$ for any $s \in [l]$ and $\mathbf{z} \in \Delta^p$ in several important cases: (0) in the absence of a loss; (1) when $\mathsf{L}$ is Markovian; (2) when $\mathsf{L}$ is a *rational loss*; and (3) when $\mathsf{L}$ is the edit-distance or any other *tropical loss*.

### D.2.1   Gradient computation in the absence of a loss

In that case, it suffices to show how to compute $Z_{\mathbf{w}}' = \sum_{y \in \mathcal{Y}} e^{\mathbf{w} \cdot \Psi(x_i, y)}$ and the following term, ignoring the loss factors:

$$\mathsf{Q}_{\mathbf{w}}'(\mathbf{z}, s) = \sum_{y: \, y_{s-p+1}^s = \mathbf{z}} \prod_{t=1}^{l} e^{\mathbf{w} \cdot \widetilde{\psi}(x_i, y_{t-p+1}^t, t)}, \tag{25}$$

for all $s \in [l]$ and $\mathbf{z} \in \Delta^p$. We will show that $\mathsf{Q}_{\mathbf{w}}'(\mathbf{z}, s)$ coincides with the flow through an edge of a weighted graph we will define, which leads to an efficient computation. We will use for any $y \in \Delta^l$, the convention $y_s = \varepsilon$ if $s \leq 0$. Now, let $\mathcal{A}$ be the weighted finite automaton (WFA) with the following set of states:

$$Q_{\mathcal{A}} = \left\{ (y_{t-p+1}^t, t) \colon y \in \Delta^l, t = 0, \ldots, l \right\},$$

with $I_{\mathcal{A}} = (\varepsilon, 0)$ its single initial state, $F_{\mathcal{A}} = \{ (y_{l-p+1}^l, l) \colon y \in \Delta^l \}$ its set of final states, and a transition from state $(y_{t-p+1}^{t-1}, t-1)$ to state $(y_{t-p+2}^{t-1} b, t)$ with label $b$ and weight $\omega(y_{t-p+1}^{t-1} b, t) = e^{\mathbf{w} \cdot \widetilde{\psi}(x_i, y_{t-p+1}^{t-1} b, t)}$, that is the following set of transitions:

$$E_{\mathcal{A}} = \left\{ \left( (y_{t-p+1}^{t-1}, t-1), b, \omega(y_{t-p+1}^{t-1} b, t), (y_{t-p+2}^{t-1} b, t) \right) \colon y \in \Delta^l, b \in \Delta, t \in [l] \right\}.$$

Figure 3: Illustration of WFA $\mathcal{A}$ for $p = 2$.

Figure 3 illustrates this construction in the case $p = 2$. The WFA $\mathcal{A}$ is deterministic by construction. The weight of a path in $\mathcal{A}$ is obtained by multiplying the weights of its constituent transitions. In view of that, $\mathsf{Q}'_{\mathbf{w}}(\mathbf{z}, s)$ can be seen as the sum of the weights of all paths in $\mathcal{A}$ going through the transition from state $(\mathbf{z}_1^{p-1}, s-1)$ to $(\mathbf{z}_2^p, s)$ with label $z_p$.

For any state $(y_{t-p+1}^t, t) \in Q_{\mathcal{A}}$, let $\alpha((y_{t-p+1}^t, t))$ denote the sum of the weights of all paths in $\mathcal{A}$ from $I_{\mathcal{A}}$ to $(y_{t-p+1}^t, t)$ and $\beta((y_{t-p+1}^t, t))$ the sum of the weights of all paths from $(y_{t-p+1}^t, t)$ to a final state. Then, $\mathsf{Q}'_{\mathbf{w}}(\mathbf{z}, s)$ is given by

$$\mathsf{Q}'_{\mathbf{w}}(\mathbf{z}, s) = \alpha\big((\mathbf{z}_1^{p-1}, s-1)\big) \times \omega(\mathbf{z}, s) \times \beta\big((\mathbf{z}_2^p, s)\big).$$

Note also that $Z'_{\mathbf{w}}$ is simply the sum of the weights of all paths in $\mathcal{A}$, that is $Z'_{\mathbf{w}} = \beta((\varepsilon, 0))$.

Since $\mathcal{A}$ is acyclic, $\alpha$ and $\beta$ can be computed for all states in linear time in the size of $\mathcal{A}$ using a single-source shortest-distance algorithm over the $(+, \times)$ semiring or the so-called forward-backward algorithm. Thus, since $\mathcal{A}$ admits $O(l|\Delta|^p)$ transitions, we can compute all of the quantities $\mathsf{Q}'_{\mathbf{w}}(\mathbf{z}, s)$, $s \in [l]$ and $z \in \Delta^p$ and $Z'_{\mathbf{w}}$, in time $O(l|\Delta|^p)$.

#### D.2.2 Gradient computation with a Markovian loss

We will say that a *loss function* $\mathsf{L}$ *is Markovian* if it admits a decomposition similar to the features, that is for all $y, y' \in \mathcal{Y}$,

$$\mathsf{L}(y, y') = \sum_{t=1}^{l} \mathsf{L}_t(y_{t-p+1}^t, {y'}_{t-p+1}^t).$$

In that case, we can absorb the losses in the transition weights and define new transition weights $\omega'$ as follows:

$$\omega'(t, y_{t-p+1}^{t-1} b) = e^{\mathsf{L}_t(y_{t-p+1}^{t-1} b, (y_i)_{t-p+1}^{t-1} b)} \omega(y_{t-p+1}^{t-1} b, t).$$

Using the resulting WFA $\mathcal{A}'$ and precisely the same techniques as those described in the previous section, we can compute all $\mathsf{Q}_{\mathbf{w}}(\mathbf{z}, s)$ in time $O(l|\Delta|^p)$. In particular, we can compute efficiently these quantities in the case of the Hamming loss which is a Markovian loss for $p = 1$.

### D.3 Efficient gradient computation for VStructBoost

In this section, we briefly describe the gradient computation for VStructBoost, which follows along similar lines as the discussion for VCRF.

Fix $i \in [m]$ and let $F_i$ denote the contribution of point $x_i$ to the empirical loss in VStructBoost. Using the equality $\mathsf{L}(y_i, y_i) = 0$, $F_i$ can be rewritten as

$$F_i(\mathbf{w}) = \frac{1}{m} \sum_{y \neq y_i} \mathsf{L}(y, y_i) e^{-\mathbf{w} \cdot \delta \Psi(x_i, y_i, y)} = \frac{1}{m} e^{-\mathbf{w} \cdot \Psi(x_i, y_i)} \sum_{y \in \Delta^l} \mathsf{L}(y, y_i) e^{\mathbf{w} \cdot \Psi(x_i, y)}.$$

The gradient of $F_i$ can therefore be expressed as follows:

$$\nabla F_i(\mathbf{w}) = \frac{1}{m} e^{-\mathbf{w} \cdot \Psi(x_i, y_i)} \sum_{y \in \Delta^l} \mathsf{L}(y, y_i) e^{\mathbf{w} \cdot \Psi(x_i, y)} \Psi(x_i, y) \qquad (26)$$

$$- \frac{1}{m} e^{-\mathbf{w} \cdot \Psi(x_i, y_i)} \Psi(x_i, y_i) \sum_{y \in \Delta^l} \mathsf{L}(y, y_i) e^{\mathbf{w} \cdot \Psi(x_i, y)}.$$

Figure 4: Illustration of the WFA $\mathcal{A}'$ for $p = 2$.

Efficient computation of these terms is not straightforward, since the sums run over exponentially many sequences $y$. However, by leveraging the Markovian property of the features, we can reduce the calculation to flow computations over a weighted directed graph, in a manner analogous to what we demonstrated for VCRF.

### D.4 Inference

In this section, we describe an efficient algorithm for inference when using Markovian features. The algorithm consists of a standard single-source shortest-path algorithm applied to a WFA $\mathcal{A}'$ differs from the WFA $\mathcal{A}$ only by the weight of each transition, defined as follows:

$$E_{\mathcal{A}'} = \left\{ \left( (\bar{y}_{t-p+1}^{t-1}, t-1), b, \mathbf{w} \cdot \widetilde{\psi}(x, y_{t-p+1}^{t-1} b, t), (\bar{y}_{t-p+2}^{t-1} b, t) \right) : y \in \Delta^l, b \in \Delta, t \in [l] \right\}.$$

Furthermore, here, the weight of a path is obtained by adding the weights of its constituent transitions. Figure 4 shows $\mathcal{A}'$ in the special case of $p = 2$. By construction, the weight of the unique accepting path in $\mathcal{A}'$ labeled with $y \in \Delta^l$ is $\sum_{t=1}^{l} \mathbf{w} \cdot \widetilde{\psi}(x, y_{t-p+1}^{t-1} b, t) = \mathbf{w} \cdot \mathbf{\Psi}(x, y)$.

Thus, the label of the single-source shortest path, $\operatorname{argmin}_{y \in \Delta^l} \mathbf{w} \cdot \mathbf{\Psi}(x, y)$, is the desired predicted label. Since $\mathcal{A}'$ is acyclic, the running-time complexity of the algorithm is linear in the size of $\mathcal{A}'$, that is $O(l|\Delta|^l)$.

# Appendix References

N. B. Atalay, K. Oflazer, and B. Say. The annotation process in the turkish treebank. In *LINC*, 2003.

K. Gimpel, N. Schneider, B. O'Connor, D. Das, D. Mills, J. Eisenstein, M. Heilman, D. Yogatama, J. Flanigan, and N. A. Smith. Part-of-speech tagging for twitter: annotation, features, and experiments. In *ACL*, 2011.

V. Koltchinskii and D. Panchenko. Empirical margin distributions and bounding the generalization error of combined classifiers. *Annals of Statistics*, 30, 2002.

V. Kuznetsov, M. Mohri, and U. Syed. Multi-class deep boosting. In *Proceedings of NIPS*, 2014.

M. Ledoux and M. Talagrand. *Probability in Banach Spaces: Isoperimetry and Processes*. Springer, 1991.

M. Mohri, A. Rostamizadeh, and A. Talwalkar. *Foundations of Machine Learning*. The MIT Press, 2012.

K. Oflazer, B. Say, D. Z. Hakkani-Tür, and G. Tür. Building a turkish treebank. In *Text, Speech and Language Technology*, volume 20, pages 261–277. Springer Netherlands, 2003.

O. Owoputi, B. O'Connor, C. Dyer, K. Gimpel, N. Schneider, and N. A. Smith. Improved part-of-speech tagging for online conversational text with word clusters. In *Proceedings of NAACL-HLT*, pages 380–390, 2013.

M. Palmer, N. Xue, F. Xia, F.-D. Chiou, Z. Jiang, and M. Chang. Chinese treebank 6.0 LDC2007T36. *Web Download*, 2007.

## Footnotes

[3] The number $S(p,n)$ of $p$-tuples $\mathbf{N}$ with $|\mathbf{N}| = n$ is known to be precisely $\binom{p+n-1}{p-1}$.

[4]To select $n$ we consider $f(n) = ce^{-nu} + \sqrt{nv}$, where $u = \rho^2/8$ and $v = \log p/m$. Taking the derivative of $f$, setting it to zero and solving for $n$, we obtain $n = -\frac{1}{2u}W_{-1}(-\frac{v}{2c^2u})$ where $W_{-1}$ is the second branch of the Lambert function (inverse of $x \mapsto xe^x$). Using the bound $-\log x \leq -W_{-1}(-x) \leq 2\log x$ leads to the following choice of $n$: $n = \left\lceil -\frac{1}{2u}\log(\frac{v}{2c^2u})\right\rceil$.

[5]Our results can be straightforwardly generalized to more complex decompositions of the form $\boldsymbol{\Psi}(\mathbf{x}, y) = \sum_{q=1}^{Q} \sum_{k=1}^{p} \boldsymbol{\Psi}_{q,k}(x, y)$.