[Reviews · NeurIPS 2016]

Reviewer 1

Summary

In this paper, a new data-dependent learning guarantee for structured prediction is introduced that is applicable for many loss functions, hypothesis sets, and different factor graph decompositions. The authors first present a new complexity measure by extending Rademacher complexity. Then, they introduce the new learning bounds for structured prediction based on their complexity measure. They show that the learning guarantees of StructSVM, Max-Margin Markov Networks, and CRF are special cases of their introduced learning guarantee in which linear functions are used for hypothesis sets. They generalize the idea of Voted Risk Minimization for structured prediction, present new learning bounds for this scenario, and introduce two new algorithms Voted CRF, and Voted StructBoost for structured prediction.

Qualitative Assessment

This paper represents an important advancement in the theoretical understanding of how structured output learning behaves. Moreover, the theory is accompanied with an algorithm that demonstrates the benefits of the new bounds. Being able to incorporate non-decomposable losses is useful, and the generality of the analysis makes it applicable to new settings where structured prediction may be important. It would be helpful for the authors to provide some intuition about the particular form of the empirical factor graph Rademacher complexity. In particular, what effects does having a Rademacher random variable for each factor, even when they overlap, have on the tightness of bounds, or the easy of estimation? Is there a guaranteed relationship between the standard Rademacher complexity? In line 188, it is mentioned that ''Note that our result shows that learning is possible even with an infinite set Y''. Is it more accurate to claim that this is true for linear hypotheses? In the equation before equation (10), instead of ''='', I think it should be ''<=''. Please clarify how equation (10) comes from the previous equation and the VRM principle. The idea of VRM is the combination of some hypothesis functions, but in equation (10), there is the sum over all the samples.

Confidence in this Review

2-Confident (read it all; understood it all reasonably well)


Reviewer 2

Summary

This paper studies the generalization error bounds for general structured prediction problems. The authors introduce the notion of factor graph complexity as an extension of Rademacher complexity and establish basic tools to estimate it. The established error bounds can recover existing results when applied to special cases. The authors also design two families of algorithms based on theoretical studies and report experimental results. The results seem to be novel and interesting.

Qualitative Assessment

The theoretical deductions rely on a contraction lemma for Rademacher complexity (Lemma 5) as an extension of Slepian's lemma. Very recently, a similar result has been independently established in "A vector-contraction inequality for Rademacher complexities" by Andreas Maurer. The idea of the proof is also similar. It seems to be better to mention this in the paper. In the specific case with Markov networks, the authors mention that "This bound can be further improved by eliminating the dependency on k using an extension of our contraction Lemma 5". In the specific case with multi-class classification, the authors mention that "the dependency on the number of classes can be further improved." I can not follow clearly these sentences. It would be helpful if the authors can explain this more clearly. It seems that the general results developed here only slightly improves the existing results in the two specific cases considered here (Markov networks and multi-class classification). Is there a specific case where the improvement is significant? Minor comments: second paragraph of section 3.3: "and and" second paragraph of section 3.3: it seems that the notation q in Taskar et al [2003]'s bound is not defined the equation above eq (10): the identity should be an inequality? the first identity in the last equation of page 12 should be an inequality? the last three y\in\mathcal{Y} in page 14 should be y\in\mathcal{Y}_f the term r_{2,\infty} in the last second equation of the proof of Corollary 10 should be r_{2,\infty}^2?

Confidence in this Review

2-Confident (read it all; understood it all reasonably well)


Reviewer 3

Summary

This paper studies generalization bounds for structured prediction. In line with previous work, the authors adopt a structured prediction model based on a scoring function. In particular they allow for such function to factorize in terms of scoring function that account only for a smaller subset of the output coordinates. A variation of the classical Rademacher complexity is proposed, based on this formulation and generalization bounds on the generalization error are discussed. The authors draw connections with previous work and then present an optimization algorithm to approach structured prediction problems.

Qualitative Assessment

The paper is well written and motivated. In particular the problem considered is relevant. On the downside there are some issues related to the interpretability of the presented results: - In Theorem 1 the generalization error is bounded in terms of the additive or multiplicative empirical margin losses. However their formulation at Eq. (5) and (6) is hard to interpret and would benefit from a comment. This is problematic since it is not clear how these quantities are related to the algorithmic approaches discussed in Sec. 5. In particular it is not clear how the approaches described in Sec. 5 are related to minimizing the additive or multiplicative empirical margin loss. [EDIT]: I thank the authors' for clarifying the role of the additive and multiplicative empirical margin loss in perspective of the algorithms proposed in Sec. 5. I feel the - Theorem 2 is quite hard to parse. In particular the role of sparsity factor s and its dependence on the maximum cardinality of each factor is not clear and deserves a deeper analysis. Indeed, in the worse case scenario it is bounded by m max_i |F_i| d_i, but in the case of NLP described in Line 192, max_i |F_i| = +\infty if I understood correctly the model (since sentences can have arbitrary length). This means that in the case of infinite dimensional output spaces only factor graphs with uniformly bounded cardinality can be used. This limit the applicability of the result and should be discussed. [EDIT]: the authors reply did not clarify my doubts. In particular, when Y is infinite: 1) why a factor graph F should have finite cardinality? and 2) the dependency of max_i |F_i| from the number of examples "m" could dramatically affect the bound. However, most structured prediction settings assume Y of finite cardinality so this is a second-order issue. - Section 3.3 draws connection with previous work. While it is interesting and relevant to understand how previous results can be cast in the setting considered, the discussion is difficult to follow. For instance, in the case of the multi-class classification the paper does not provide a direct formulation of the problem within their notation and therefore it is not clear how to interpret the theoretical results in this setting. Moreover, to connect to previous work the authors use results (e.g. Lemma 6), which are reported in the appendices and not directly connected to the main results of this paper (Theorem 1 and 2). This makes the discussion harder to follow. - Sec. 5: as mentioned above, one problem of the discussion in Sec. 5 is that it lacks a direct connection with the theoretical results presented in this work. A further issue is that the authors do not comment on the relation between the ideal empirical (non-convex) problem and the convex surrogates that are reviewed/proposed. In particular it would be interesting to understand how tight the relation of the VCRF method is with specific losses. For instance in the case of multi-class classification it is known that StructSVM - which belongs to the general framework proposed in Eq. (9) - is not consistent. Minor notes: - Line 158: the authors state that the empirical factor graph Rademacher complexity is sharply concentrated around its expectation but I was not able to find a proof of this in the supplementary material. - Line 168: "linear functions", did the authors mean "linear spaces"? - Line 296 - 297: the authors used \delta \psi(x,y,y') which was never defined.

Confidence in this Review

2-Confident (read it all; understood it all reasonably well)


Reviewer 4

Summary

The authors present a general framework for bounding the generalization errors in structured prediction tasks. The results are broadly applicable for tasks where the score function decomposes as a sum of components, where each component depends on a small number of output nodes characterized by a factor graph.

Qualitative Assessment

The generalization bounds are derived in terms of a new Rademacher complexity measure that explicitly captures the factor graph structure. The results are extended for ensemble learning using voted risk minimization. The applicability of the theorem is illustrated for various structured prediction task including linear estimators for multiclass prediction, CRFs and pairwise markov networks. The main theorem is applicable for strictly generalized framework compared to existing results in structure prediction. I have a high level question regarding the empirical factor graph Rademacher complexity expression in line 108: In general {Y_f:f\in F_i} are not independent. In the current definition, a common node could potentially be assigned conflicting labels. I have not checked the supplementary material, so could the authors clarify that this dependency handled in the analysis? [EDIT]: I thank the authors for their response. I went through the appendix in greater detail had a few suggestions. Lemma 5 (Contraction inequality for vector valued functions and Rademacher variables) is a non-trivial result and key component in the proof and of independent interest to a wider audience (this could be highlighted in the main paper). Also, a similar result for Gaussian complexity is dealt with in Theorem 14 in the classical paper by Bartlett and Mendelson, "Rademacher and Gaussian Complexities: Risk Bounds and Structural Results". The proof of the main theorems themselves are straightforward given Lemma 5, and could potentially be tightened (not completely sure though). Specifically, I am a little bothered by d_i factor in line 183 which can sometimes grow exponentially, eg in multi-label setting, d_i=max_f |Y_f|=2^(size of factor with most labels). This also relates to Assigned_Reviewer_6's question, as for infinite dimensional Y, if we assume |F_i| to be finite, then d_i will definitely be infinity.

Confidence in this Review

2-Confident (read it all; understood it all reasonably well)


Reviewer 5

Summary

- Score functions for structured predictions are traditionally decomposed into factors. - By introducing a new data dependent measure of complexity, the empirical factor graph Radamacher complexity, the authors are able to tackle the difficult task of providing a general theoretical analysis of structured prediction. - Furthermore, they show that the notion of specific factor complexity can be combined with the recently introduced Voted Risk Minimisation (VRM) scheme. This results in a new set of algorithms that allow for the possible inclusion of complex hypothesis classes, without having to worry about overfitting i.e. trades of hypothesis class complexity with fit. - Experiments are presented which show the VRM scheme coupled with conditional random fields (CRFs), outperforms the vanilla CRF methods on a variety of structured output problems.

Qualitative Assessment

Although the focus is on the theoretical contributions, I still would have liked to see the outcomes of the experiments in the main body of the paper rather than appendix. Apart from that a thoroughly enjoyable read and a great contribution to the field of structured prediction.

Confidence in this Review

2-Confident (read it all; understood it all reasonably well)